# Aircraft Observations of the Chemical Composition and Aging of Aerosol in the Manaus Urban Plume during GoAmazon 2014/5

John E. Shilling[1], Mikhail S. Pekour[1], Edward C. Fortner[2], Paulo Artaxo[3], Suzane de Sá[4], John M. Hubbe[1], Karla M. Longo[5], Luiz A.T. Machado[6], Scot T. Martin[4,7], Stephen R. Springston[8], Jason Tomlinson[1], Jian Wang[8]

[1]Atmospheric Sciences and Global Change Division, Pacific Northwest National Laboratory, Richland, Washington, USA
[2]Center for Aerosol and Cloud Chemistry, Aerodyne Research, Billerica, Massachusetts, USA
[3]Institute of Physics, University of Sao Paulo, Sao Paulo, Brazil
[4]John A. Paulson School of Engineering and Applied Sciences and Department of Earth and Planetary Sciences, Harvard University, Cambridge, Massachusetts, USA
[5]University Space Research Association/Goddard Earth Sciences Technology and Research (USRA/GESTAR), National Aeronautics and Space Administration, Goddard Space Flight Center, Columbia, Maryland, USA
[6]Centro de Previsao de Tempo e Estudos Climaticos – Instituto Nacional de Pesquisas Espaciais, Sao Jose dos Campos, Brazil
[7]Department of Earth and Planetary Sciences, Harvard University, Cambridge, Massachusetts, USA
[8]Environmental and Climate Sciences Department, Brookhaven National Laboratory, Upton, New York, USA

*Correspondence to*: John E. Shilling (john.shilling@pnnl.gov)

**Abstract.** The Green Ocean Amazon (GoAmazon 2014/5) campaign, conducted from January 2014 – December 2015 in the vicinity of Manaus, Brazil, was designed to study the aerosol lifecycle and aerosol-cloud interactions in both pristine and anthropogenically-influenced conditions. As part of this campaign, the U.S. Department of Energy (DOE) G-1 research aircraft was deployed from February 17 – March 25, 2014 (wet season) and September 6 – October 5, 2014 (dry season) to investigate aerosol and cloud properties aloft. Here, we present results from the G-1 deployments focusing on measurements of the aerosol chemical composition and discussion of aerosol sources and secondary organic aerosol formation and aging.

In the first portion of the manuscript, we provide an overview of the data and compare and contrast the data from the wet and dry season. Organic aerosol (OA) dominates the deployment-averaged chemical composition, comprising 80% of the non-refractory $PM_1$ aerosol mass with sulfate comprising 14%, nitrate 2%, and ammonium 4%. This product distribution was unchanged between seasons, despite the fact that total aerosol loading was significantly higher in the dry season and that regional and local biomass burning was a significant source of OA mass in the dry, but not wet, season. However, the OA was more oxidized in the dry season, with the median of the mean carbon oxidation state increasing from -0.45 in the wet season to -0.02 in the dry season.

In the second portion of the manuscript, we discuss the evolution of the Manaus plume, focusing on March 13, 2014, one of the exemplar days in the wet season. On this flight, we observe a clear increase in OA concentrations in the Manaus plume relative to the background. As the plume is transported downwind and ages, we observe dynamic changes in the OA. The mean carbon oxidation state of the OA increases from -0.6 to -0.45 during the 4-5 hours of photochemical aging. Hydrocarbon-like organic aerosol (HOA) mass is lost with $\Delta HOA/\Delta CO$ values decreasing from 17.6 $\mu g/m^3$ $ppmv^{-1}$ over Manaus to 10.6

μg/m$^3$ ppmv$^{-1}$ 95 km downwind. Loss of HOA is balanced out by formation of oxygenated organic aerosol (OOA) with ΔOOA/ΔCO increasing from 9.2 to 23.1 μg/m$^3$ ppmv$^{-1}$. Because HOA loss is balanced by OOA formation, we observe little change in the net Δorg/ΔCO values; Δorg/ΔCO averages 31 μg/m$^3$ ppmv$^{-1}$ and does not increase with aging. Analysis of the Manaus plume evolution using data from two additional flights in the wet season showed similar trends in Δorg/ΔCO as the

March 13 flight; Δorg/ΔCO values averaged 34 μg/m$^3$ ppmv$^{-1}$ and showed little change over 4-6.5 hours of aging. Our observation of constant Δorg/ΔCO are in contrast to literature studies of the outflow of several North American cities, which report significant increases in Δorg/ΔCO for the first day of plume aging. These observations suggest that SOA formation in the Manaus plume occurs, at least in part, by a different mechanism than observed in urban outflow plumes in most other literature studies. Constant Δorg/ΔCO with plume aging has been observed in many biomass burning plumes, but we are

unaware of reports of fresh urban emissions aging in this manner. These observations show that urban pollution emitted from Manaus in the wet season forms much less particulate downwind than urban pollution emitted from North American cities.

## 1 Introduction

Aerosol particles have important impacts on visibility, human health, and the Earth's energy balance and water cycle. The impact of aerosol particles on radiation balance, in particular their impact on cloud properties and lifetimes, continues to be a

significant source of uncertainty for global climate models (Intergovernmental Panel on Climate, 2014). An extensive series of field studies has shown that a large fraction of the total non-refractory aerosol mass is organic aerosol (OA) and that a large fraction of this OA mass forms in the atmosphere when organic compounds in the gas phase are oxidized and subsequently condense as secondary organic aerosol (SOA) (Zhang et al., 2007;Jimenez et al., 2009). Because SOA is such a large fraction of the aerosol mass, condensation of SOA is critical to growing nucleation-mode particles, which are initially too small to

serve as cloud condensation nuclei (CCN), to sizes that are capable of forming cloud droplets (Ehn et al., 2014;Riipinen et al., 2011;Pierce et al., 2012), though a recent study showed that unique conditions in the Amazon allowed particles smaller than 50 nm to act as CCN during GoAmazon 2014/5 (Fan et al., 2018). Thus, accurate descriptions of SOA condensation and aerosol growth kinetics are crucial to accurately predicting aerosol size distributions and therefore CCN number concentrations and aerosol optical properties, both of which are required for accurately predicting the impact of aerosols on climate (Scott et

al., 2015;Riipinen et al., 2012;Zaveri et al., 2014).

For several years, there has been an interest in studying the Lagrangian (i.e., within the same air parcel) evolution of organic aerosol from the emissions of urban centers. Field studies have investigated the evolution of pollution plumes   by arranging fixed observation sites at different distances downwind of a city along the direction of the prevailing wind or by tracking the plume with a mobile platform, such as an aircraft. Larger campaigns may employ both strategies. Most studies of this type are

best described as pseudo-Lagrangian as repeatedly sampling the same air parcel is difficult with mobile platforms and impossible with fixed sites. Dilution and mixing of the air parcel with background air also alter the plume composition. To account for atmospheric dilution and spatial and temporal variability in emissions, studies often utilize the ratio of excess OA

to that of an inert tracer as a metric for evaluating OA formation in a plume (Kleinman et al., 2008;Sullivan and Weber, 2006;Takegawa et al., 2006;de Gouw et al., 2005). CO is a common choice for the inert tracer because it is emitted during combustion and other anthropogenic processes, is significantly enhanced in urban plumes relative to the background, and is routinely and robustly measured. Measurements of $\Delta$org/$\Delta$CO in aged urban outflow have spanned a range from 47 $\mu$g/m$^3$

ppmv$^{-1}$ in the NE US (de Gouw et al., 2008), to 62-80 $\mu$g/m$^3$ ppmv$^{-1}$ in Mexico City (DeCarlo et al., 2008;Kleinman et al., 2008), to 100 $\mu$g/m$^3$ ppmv$^{-1}$ in the Po Valley, Italy (Crosier et al., 2007), to 44 – 197 $\mu$g/m$^3$ ppmv$^{-1}$ in Sacramento, CA (Shilling et al., 2013;Setyan et al., 2012), to 97-133 $\mu$g/m$^3$ ppmv$^{-1}$ in Paris (Freney et al., 2014). The same studies have found that $\Delta$org/$\Delta$CO increases roughly linearly with air mass age for the first 1 day of aging and levels off after approximately 2 days (de Gouw and Jimenez, 2009;DeCarlo et al., 2010;Kleinman et al., 2008;Takegawa et al., 2006;Sullivan et al., 2006;Freney et

al., 2014). Observations suggest that changes in $\Delta$org/$\Delta$CO begin soon after emission. For example, measurements during the MILAGRO campaign showed that $\Delta$org/$\Delta$CO increased from 10-35 $\mu$g/m$^3$ ppmv$^{-1}$ for fresh emissions to 70-80 $\mu$g/m$^3$ ppmv$^{-1}$ after approximately one day of photochemical aging (DeCarlo et al., 2010;Kleinman et al., 2008). Changes of similar magnitude over similar aging times have been measure in the urban outflow of: Tokyo (Takegawa et al., 2006), the SE USA (Sullivan et al., 2006), Paris (Freney et al., 2014), and the NE USA (de Gouw et al., 2005).

Measurements have also suggested that organic aerosol production may be enhanced when urban emissions interact with biogenic emissions. Organic aerosol concentrations have shown strong correlation with CO and other tracers of anthropogenic emissions (de Gouw et al., 2005;Volkamer et al., 2006;Weber et al., 2007;Sullivan et al., 2006). Furthermore, satellite observations suggest that the spatial and seasonal patterns of aerosol optical depth coincide with biogenic emissions (Goldstein et al., 2009). At the same time, radiocarbon dating has shown that a large fraction of the carbon in the aerosol phase is modern

and that the modern carbon fraction increased with increasing distance from urban centers (Weber et al., 2007;Schichtel et al., 2008). During the CARES campaign near Sacramento, CA, aircraft observations showed that $\Delta$org/$\Delta$CO measurements were 35-44 $\mu$g/m$^3$ ppmv$^{-1}$ when anthropogenic emissions evolved in the absence of strong biogenic emissions and 77-157 $\mu$g/m$^3$ ppmv$^{-1}$ when the urban plume interacted with regions of strong biogenic emissions (Shilling et al., 2013). Measurements from a ground site during CARES report average $\Delta$org/$\Delta$CO values of 36 $\mu$g/m$^3$ ppmv$^{-1}$ in the Sacramento plume during periods of

low biogenic emissions and 97 $\mu$g/m$^3$ ppmv$^{-1}$ during periods of high biogenic emissions (Setyan et al., 2012). Several additional field studies report enhancements of SOA through interactions of anthropogenic and biogenic emissions, but may not employ the $\Delta$org/$\Delta$CO metric (Zhou et al., 2016;Xu et al., 2015;Ng et al., 2017;Bean et al., 2016). The potential mechanisms responsible for these enhancements are known in some cases but remain uncertain in others and have been outlined in recent review articles (Shrivastava et al., 2017;Ng et al., 2017;Glasius and Goldstein, 2016;Hoyle et al., 2011).

To date, most studies of the evolution of urban plumes have been conducted in the Northern Hemisphere. Far fewer studies have been performed in the Southern Hemisphere where there is less landmass, lower population, and therefore lower background concentrations of anthropogenic pollutants (Andreae, 2007). The Amazon tropical forest, in particular, is an important ecosystem in which background levels of anthropogenic pollution may sometimes reach levels characteristic of pre-industrial conditions but is becoming increasingly impacted by industrialization and anthropogenic pollution (Martin et al.,

2017;Martin et al., 2010b;Andreae et al., 2015). Fuzzi et al. (2007) described chemical analysis of filter samples collected in Rondonia, Brazil (~825 km SW of Manaus, Brazil) from September to November 2002. They found that water-soluble organic species were the main contributors to submicron particles (90%) with the balance consisting of soluble inorganic ions (Fuzzi et al., 2007). Allen et al. (2014) discuss measurements from flights investigating biogenic processes during the South American

Biomass Burning Analysis (SAMBBA) aircraft campaign, based out of Rondonia, Brazil, They report that concentrations of SOA derived from isoprene epoxydiols (IEPOX) were highest in the presence of acidic seed, low $NO_x$ concentrations, and high RH (Allan et al., 2014). The AMAZE-08 campaign was conducted in February and March of 2008 and Chen et al. (2009, 2015) reported on chemical composition measurements made with an AMS located at a tower site 60 km NNW of Manaus, Brazil (Martin et al., 2010a;Chen et al., 2009;Chen et al., 2015). They found that organics accounted for more than 80% of the

non-refractory aerosol mass, with the organic mass often dominated by SOA, and that the aerosol was acidic in composition (Chen et al., 2015). It is important to note that Chen et al. (2009, 2015) excluded data impacted by Manaus emissions from their analysis. de Sá et al. (2017) investigated the production of IEPOX SOA downwind of Manus during GoAmazon2014/5 and found decreased concentrations of IEPOX-SOA in the plume relative to the background which they attribute to suppression of IEPOX formation by elevated NO concentrations in the plume. Kuhn et al. (2010) describe aircraft-based aerosol

measurements in the Manaus plume as it was transported downwind. They report increases in ΔCCN/ΔCO as the plume aged, which they attribute to condensation of both organic and inorganic mass, but did not measure the aerosol chemical composition (Kuhn et al., 2010). Additional aircraft campaigns conducted in the Amazon often focused on biomass burning and the impact of biomass burning aerosol on cloud properties (Morgan et al., 2013;Yokelson et al., 2007;Andreae et al., 2012).

The Green Ocean Amazon (GoAmazon 2014/5) campaign was conducted in the vicinity of Manaus, Brazil from January 2014

– December 2015. The goal of GoAmazon 2014/5 was to investigate the interaction of Manaus urban emissions with the surrounding pristine Amazon basin and the subsequent impact of these emission on cloud formation and properties (Martin et al., 2017;Martin et al., 2016). Manaus's geography makes it ideal for this mission. Manaus, an industrial city and with a metropolitan population of more than 2 million people, is the largest city in the Amazon basin. The prevailing wind is from the east with the nearest major upwind city, Belem, approximately 1250 km away. The surrounding tropical forest emits vast

quantities of biogenic gases and aerosol. Few roads connect Manaus to the rest of Brazil and most freight and traffic from outside the city is via ship or plane. Thus, Manaus acts as a large point source of anthropogenic emissions which are transported to the surrounding Amazon basin. As part of this campaign, the DOE Gulfstream-1 (G-1) research aircraft conducted two, six-week-long missions in which it investigated the evolution of the Manaus plume as it was transported into the surrounding Amazon tropical rainforest.  The timing of these flight missions was chosen to provide a contrast between the wet and dry

seasons (Martin et al., 2016). In the wet season, back trajectory analysis indicated that the Manaus region was typically under the influence of air originating from the North Atlantic Ocean (Martin et al., 2016). Regular, organized mesoscale convective systems triggered by sea breeze circulation brought widespread, moderate rate precipitation to the region (Giangrande et al., 2017;Machado et al., 2018;Burleyson et al., 2016). The high level of rainfall and moisture in the wet season inhibited biomass burning and a low number of fires were observed (Martin et al., 2016;Machado et al., 2018). Under these conditions, the

Amazon basin is one of the cleanest continental regions on Earth onto which the Manaus plume represents a significant perturbation (Martin et al., 2010b;Artaxo et al., 2013). In the dry season, back trajectory analysis indicate air masses originate from the Southern Hemisphere and travel up the Amazon River transporting pollution from the northern coastal cities (Martin et al., 2016). In addition, recirculation events transported air from the southern Amazon basin into the Manaus region (Martin et al., 2016). Intense biomass burning fires in the central and southern portion of the Amazon basin and in central Africa were observed in the dry season and a portion of these emissions were transported to the Manaus region (Martin et al., 2016;Artaxo et al., 2013;Martin et al., 2010b). In the dry season, more intense but less frequent convection produced approximately one quarter of the total rainfall observed in the wet season. As a result of the combination of transport and precipitation frequency, the Manaus region is significantly more polluted in the dry season.

In this manuscript we report on measurements from instruments deployed on the G-1 focusing primarily on measurements of aerosol species and trace gases that impact the aerosol lifecycle. In the first part of the manuscript, we provide an overview of aerosol and VOC measurements and compare and contrast the wet and dry season data. In the second portion of the manuscript, we examine, in detail, the first 4 – 6.5 hours of photochemical aging of the Manaus plume as it is transported into the surrounding tropical forest and interacts with biogenic emissions.

## 2 Experimental

### 2.1 G-1 Flight Strategy

The GoAmazon 2014/5 campaign was conducted in the vicinity of Manaus, Brazil from January 1, 2014 through December 31, 2015 (Martin et al., 2016). During this period, DOE's G-1 research aircraft, based out of the Manaus International Airport, was deployed for two periods during which time it sampled the Manaus urban plume as it was transported downwind over the Amazon rainforest. The first aircraft deployment period occurred during the wet season from February 15 - March 26, 2014 while the second occurred in the dry season from September 1 - October 10, 2014. Sixteen research flights were conducted in the wet season and 19 were conducted in the dry season. Table 1 lists takeoff and landing times, the altitude of level flight legs, and meteorological parameters measured by the G-1 instrumentation shortly after takeoff on each flight. In general, flight plans were focused on successive perpendicular plume crossings spaced approximately equally at intervals of 24 km downwind of the city. A representative flight path for March 13, 2014, an exemplar day in the wet season with few clouds, is shown in Figure 1. The paths for all flights during the two deployments are shown, segregated by altitude, in Martin et al. (2016) and therefore are not reproduced here. On March 13, the first leg approximately bisected the city along a NW/SE line, the second leg passed near a highly instrumented ground site directly across the Amazon river from Manaus (T2), the fourth leg passed over a second highly instrumented site located NE of Manacaparu (T3), with an additional leg between the T2 and T3 sites, and a final leg downwind of the T3 site. The first pass through the pattern was generally at an altitude of 500-700 m above ground level with a second pass at a higher altitude and often, though not always, focused on sampling clouds. The flight

pattern discussed above and shown in Figure 1 was rotated to align with the prevailing wind direction. Flights generally departed in the late morning or early afternoon and lasted between 3 and 3.5 hours (Table 1).

## 2.2 Instrumentation

An Aerodyne High-Resolution Time-of-Flight Aerosol Mass Spectrometer (abbreviated as AMS hereafter) was deployed on the G-1 to measure aerosol chemical composition (Jayne et al., 2000;DeCarlo et al., 2006). The AMS operated only in the standard "V"- MS mode (the particle sizing mode was not used) with a 13s data averaging interval and equal chopper open and closed periods of 3 seconds. Before, during, and after flights, the AMS-sampled air was periodically diverted through a HEPA filter to remove particulates and these filter periods were used to account for gas-phase interferences with isobaric particulate signals. Based on the standard deviation of these blank measurements ($3\sigma$) as described in the literature (DeCarlo et al., 2006), the detection limit of the AMS at the 13s sampling interval were 0.12, 0.01, 0.014, and 0.005 $\mu g/m^3$ for organics, sulfate, nitrate, and ammonium respectively. All aerosol instruments, including the AMS, sampled from a common, double-diffuser isokinetic inlet. Flow for the AMS was sub-sampled from the center of the isokinetic inlet and dried to RH < 40% by passage through a 1/2" ID PermaPure Nafion membrane. A constant pressure inlet operating at approximately 620 hPa was used to maintain a constant volumetric flow to the AMS up to altitudes of approximately 3900 m (Bahreini et al., 2008). Because the AMS was powered off between flights, a correction based on the real-time $N_2^+$ signal was applied to all data to account for drifting sensitivity as the instrument warmed up during flights. The AMS was regularly calibrated in the field using monodisperse ammonium nitrate particles quantified with a TSI condensation particle counter (CPC). Data was analyzed in Igor Pro (v6.37) using the high-resolution analysis package (Squirrel v1.55H, PIKA v1.44H) and techniques described in the literature (Canagaratna et al., 2015;Kroll et al., 2011;Aiken et al., 2007;Allan et al., 2003;Jimenez et al., 2003). All AMS data in the manuscript have been processed using the high-resolution data analysis routine. The O:C and H:C values reported here use the updated calibrations described in Canagaratna et al. (2015). All AMS data are normalized to the laboratory calibration conditions of 23 °C 1013 hPa. Primary Matrix Factorization (PMF) analysis was performed on the wet-season dataset by combining all flight data into a single experiment using the PMF Evaluation Tool (v 2.06) and the PFM2 algorithm (v 4.2) (Ulbrich et al., 2009;Paatero, 1997;Paatero and Tapper, 1994).

An Ionicon quadrupole high-sensitivity Proton Transfer Reaction Mass Spectrometer (PTR-MS) was used to measure selected gas-phase volatile organic carbon (VOC) concentrations (Lindinger et al., 1998). The PTR-MS was run in the ion monitoring mode in which signals of a limited number of pre-selected of m/z values are sequentially measured with one measurement cycle taking 3.5 s. Averaging time at each m/z in the series varied depending on the sensitivity of the instrument to that species, the expected concentration, and the background, but generally varied between 0.2 and 0.5 s. Isoprene at m/z 69 was sampled multiple times during the cycle to enable flux analysis (Gu et al., 2017). Drift tube temperature, pressure and voltage were held at 60 °C, 2.22 hPa, and 600 V, respectively resulting in an electric field to gas density (E/N) ratio of 134 Td (1 Td =$1\times10^{-17}$ $cm^2V^{-1}s^{-1}$). The PTR-MS sampled air through a dedicated forward-facing inlet that consisted of approximately 6" of 1/4"

OD stainless steel followed by approximately 46" of 1/4" Teflon tubing, and 36" of 1/16" OD PEEK tubing. The flow through the Teflon tubing was 600 ccm with 300 ccm subsampled through the PEEK tubing for introduction into the PTR-MS. To assess the PTR-MS background, air was periodically diverted through a stainless steel tube filled with Shimadzu Pt catalyst heated to 600 °C, which removes VOCs from the airstream without perturbing relative humidity. The catalyst efficiency was

tested during the campaign by comparing signal from air containing VOCs passed through the catalyst with signal from VOC-free air. The PTR-MS was calibrated by introducing known concentrations of calibration gases into the instrument with variable dilution by VOC-free air. The calibration tank VOC concentrations were determined gravimetrically and verified using GC analysis by the manufacturer (AiR Environmental, Inc).

Ozone was measured with a Thermo Scientific Model 49i ozone analyzer based on measurement of UV absorption at 254 nm.

The instrument was regularly calibrated in-flight by displacement of known quantities of ozone and zeroed in flight using ozone-scrubbed ambient air. CO was measured using a Los Gatos Research $CO/N_2O/H_2O$ analyzer that is based on cavity-enhanced near-IR absorption and was also calibrated regularly in-flight. Additional information on the instrumentation deployed on the G-1 can be found in GoAmazon 2014/5 overview manuscript (Martin et al., 2016).

## 3 Results and Discussion

3.1 Overview of G-1 Aerosol Data and Comparison of Wet and Dry SeasonsAs discussed in the introduction, the different precipitation and mesoscale transport patterns in the wet and the dry season are expected to produce considerably different concentrations of pollutants in the Amazon basin and measurements confirmed this expectation. Figure 2 and Table 2 summarize the AMS results for data collected on the G-1 flights in both the wet and the dry season. The top panel of Figure 2 shows box-and-whisker plots summarizing AMS data for organics, sulfate, nitrate, and ammonium particulate concentrations,

with one data point representing each flight (note the scale change between the wet and dry season plot). The middle panel shows the relative distribution of the chemical species on each flight. The bottom panel shows the distribution of the chemical species as a mass weighted average of all data from the respective season. Several trends are readily apparent in the data. First, it is clear that aerosol loadings were significantly higher in the dry season than in the wet season. Median organic loadings for the all wet and dry season data were 0.85 and 4.29 $\mu g/m^3$, respectively. Median sulfate, nitrate and ammonium loadings

increased from 0.19, 0.02, and 0.06 $\mu g/m^3$ in the wet season to 0.77, 0.05, and 0.25 $\mu g/m^3$ in the dry season. Mean loadings of all species were somewhat higher than the median. Mean organic, sulfate, nitrate, and ammonium loadings were 0.91, 0.16, 0.02, and 0.05 $\mu g/m^3$ in the wet season and 4.41, 0.80, 0.1, and 0.26 $\mu g/m^3$ in the dry season. In the wet season, the mass loading of all species are lower than typically observed over continental regions in the Northern Hemisphere (Zhang et al., 2007).

While this manuscript will focus on the particulate data, measurements of the volatile organic compounds provide insights into the precursors that are oxidized to form OA and help to identify the source of an air parcel. Figure 3 and Table 3 summarize the concentrations of several relevant VOCs measured on board the G-1 with the PTR-MS. Similar to the trends seen in the

aerosol mass loading data, concentrations of most VOCs measured by the PTR-MS are significantly higher in the dry season than in the wet season. Concentrations of isoprene and its oxidation products, biogenic precursors of OA, are a factor of 2-3 times higher in the dry season than the wet season. The average daily high temperatures in Manaus are 33.5°C in September (dry season) and 30.9°C in March (wet season) and isoprene emissions have been shown to scale with temperature, among

other variables (Guenther et al., 2012). Measurements of benzene, which is primarily anthropogenic in origin, were also significantly higher in the dry season. While benzene itself is unlikely to significantly contribute to SOA formation over the timescales we observe on the G-1 flights (4-6.5 hours) due to its ~5 day atmospheric lifetime (Atkinson and Arey, 2003), other unmeasured anthropogenic VOC concentrations which would contribute to more immediate OA formation may have been higher as well. The higher VOC concentrations may contribute to the higher OA concentrations measured in the dry season.

However, the PTR-MS data, in addition to satellite and remote sensing measurements (Martin et al., 2016), also show that biomass burning significantly impacted the region. Measurements of acetonitrile, whose major source is biomass burning (Yokelson et al., 2009;Yokelson et al., 2007), are 2-3 times higher in the dry season than the wet season, indicating a significant biomass burning impact in the region. Biomass burning is a known source of OA (Jolleys et al., 2012;Bond et al., 2004;Yokelson et al., 2009;Ferek et al., 1998) and would also contribute to the higher aerosol concentrations observed in the

dry season. The aerosol mass loadings we measure in the dry season are consistent with aircraft-based AMS measurements made during the South American Biomass Burning Analysis (SAMBBA) campaign at the same time of year, though several hundred kilometers to the SW in the state of Rondonia (Allan et al., 2014). Chen et al. reported AMS-measured wet-season campaign (February 7 to March 13, 2008) average organic and sulfate loadings of 0.7 and 0.15 μg/m$^3$ from a ground site that was within the GoAmazon 2014/5 flight domain, though they excluded data impacted by Manaus emissions from their analysis

(Chen et al., 2009). When comparing aircraft to ground site data, it is important to acknowledge differences in the datasets. First, the aircraft samples a different spatial domain in all three dimensions (latitude, longitude, and altitude) than a ground site. In addition, the G-1 typically sampled the boundary layer in the late morning (see Table 1 for flight times), and thus may miss peak concentrations of secondary species. Finally, the goal of the flights was often to sample the Manaus plume for the first half of each flight and clouds for the second half. The mean aerosol loadings reported here agree well with those reported

by Chen et al. (2009), considering the inherent differences in datasets discussed above. We also found good agreement when comparing the G-1 measured aerosol loadings to those measured at the T3 site when the aircraft passed overhead and at 500 – 600 m altitude (de Sá et al., 2018). Our loadings are at the low end of the range (1- 2 μg/m$^3$) reported from a ground site in Rondonia, approximately 825 km to the southwest of Manaus, though those data were more heavily influenced by biomass burning and relied on a combination of offline techniques to speciate the aerosol (Fuzzi et al., 2007). To understand potential

bias in the dataset due to altitude, we calculated statistics for data collected below 700 m, which captures the lowest altitude portion of each flight. Giagrande et al. (2017) analyzed radiosonde data from the T3 site and report average mixed layer heights of greater than 1000 m above ground level at 10:00 local time in both the wet and dry seasons. Thus, the G-1 was typically sampling in the boundary layer on the lowest flight legs and the 700 m data should represent boundary layer conditions. In the

wet season, the low altitude AMS loading statistics are generally either slightly higher or unchanged relative to the full dataset. In the dry season, the low altitude AMS loading statistics are either slightly lower relative or unchanged relative to the full dataset, likely due to transport of biomass burning from the south in elevated layers. Both the median and mean concentrations of most VOCs measured are slightly higher for the data at < 700 m altitude relative to the full dataset (Table 3). This is the

expected behavior for most VOCs measured here as they are either directly emitted at the surface (e.g., isoprene, benzene, monoterpenes) or form from oxidation of VOCs emitted at the surface (e.g., isoprene oxidation products).

The fractional contribution of each species to the total loading is nearly identical when comparing seasons, despite the large differences in aerosol absolute mass concentrations and the larger influence of biomass burning in the dry season (Martin et al., 2016). Organics dominate the chemical composition, comprising 80 % of the total with sulfate 14%, nitrate 2%, and

ammonium 4%, regardless of the season. The relative distribution of the products changes only modestly among flights (Figure 2, middle panel). Organics were the largest fraction of the total aerosol mass on each flight, varying between 72 – 83%. Sulfate was the second largest fraction of the total mass (7-20%) and the sum of sulfate and organic were nearly constant. The fractional contributions of each chemical species described herein are nearly identical to the previous reports from Chen et al. (2009, 2015). The aircraft-measured chemical composition is also nearly identical to that measured at the T3 ground site (de Sá et al.,

2018) during GoAmazon 2014/5. The agreement between the T3 and G-1 product distributions suggests that the T3 ground site is sampling air that is representative of the regional air and is not unduly biased by local emissions, at least in the aggregate. The observation that the total concentrations of all aerosol components significantly increases yet remain in the same proportions when comparing the wet and the dry season is unexpected. The more frequent and widespread precipitation events in the wet season and the lower concentrations of precursor VOCs contribute to the lower aerosol concentrations in the wet

season. If the aerosol particles are internally mixed, it would be reasonable to assume that rainfall would remove aerosol in approximately equal proportions. However, as discussed previously, aerosol sources are different between seasons and there was a much larger biomass burning influence in the dry season (Martin et al., 2016), which is a known and significant source of OA (e.g., (Jolleys et al., 2012;Bond et al., 2004;Yokelson et al., 2009;Ferek et al., 1998). Biomass burning was previously shown to emit particulate sulfate and nitrate directly, and $NO_x$, sulfuric acid, and methane sulfonic acid (MSA), precursors of

particulate nitrate and sulfate, but not in the same ratio as OA and its precursors (Yokelson et al., 2009). Thus, given the significant influence of biomass burning in the dry season, it is surprising that the chemical composition of the aerosol does not change.

Despite the similarity of the distribution of organic, sulfate and nitrate in the aerosol particles between the wet and dry seasons, the chemical composition of the organic aerosol is quite different between seasons. Figure 4 shows normalized probability

distributions for three metrics of the organic aerosol composition, the hydrogen to carbon ratio (H:C), the oxygen to carbon ratio (O:C), and the mean carbon oxidation state ($\bar{O}S_c$) segregated by season. It is clear that the organic particles are more oxidized in the dry season than the wet season. The median O:C, H:C, and $\bar{O}S_c$ in the dry season were, 0.78, 1.58, and -0.02 respectively, while they were 0.6, 1.65, and -0.45 in the wet season. While the H:C ratios change only slightly between season, the O:C and $\bar{O}S_c$ both show significant increases in the dry season. Thus, the aerosol in the dry season is significantly more

aged, consistent with aged biomass burning aerosol dominating the organic aerosol mass in the dry season and relatively fresh, locally generated organic aerosol dominating in the wet season (Martin et al., 2010b). The probability distributions during the wet season are also wider than in the dry season. We postulate that small contributions from a wider range of sources are able to influence the data to a larger degree in the wet season, when the total organic aerosol concentrations were much smaller. In contrast, during the dry season, small sources of organic mass would have a smaller impact on the organic composition due to the significant biomass burning background.

## 3.2 Evolution of Organic Aerosol in the Manaus Plume

Figure 1 shows the flight path on March 13, 2014, which is an exemplar day for observing the evolution of the Manaus plume in the wet season. Flight legs were perpendicular to the prevailing wind direction and spaced approximately 24 km apart at successively increasing distance from Manaus. The length of the legs was chosen such that measurements (e.g., CO concentration) returned to near background levels at the ends of the leg. Due to dispersion of the plume, the length of the legs generally increased downwind of Manaus, ranging from 56 km close to the city to 80 km downwind of the T3 site. The pattern was flown first at an altitude of 500 m with a second pass at 1000 m overlapping the first. This flight was marked by mostly sunny skies with no clouds intercepted by the G-1 at 500 m and only brief passage through spatially small clouds at 1000 m. There was little local or regional biomass burning during this time period. Thus, this flight represents a case with coherent transport of the Manaus plume downwind into regions of high biogenic emissions with few complicating factors, such as biomass burning or cloud processing.

Figure 5 shows a time-series of several important gas- and particle-phase species for the March 13, 2014 flight. Passage through the Manaus plume is indicated by clear and significant increases in CO, ozone, and PTR-MS m/z 79 (benzene) above the background levels. While CO may be produced from the oxidation of biogenic VOCs (Slowik et al., 2010), the intense peaks are primarily from urban emissions, given the proximity to Manaus and the correlation of CO with other known anthropogenic species, such as ozone, benzene, and toluene (not shown). Concentrations of CO (80 ppbv) and ozone (10 ppbv) are lowest at the edges of the flight legs and, although we did not sample extensively in regions that were far removed from Manaus, these concentrations are consistent with the background concentrations that have been previously reported in the region (Andreae et al., 2015;Martin et al., 2010b;Harriss et al., 1990). Isoprene concentrations (PTR-MS m/z 69) typically reach their highest levels outside of the plume and are lower inside, though isoprene concentrations of approximately 1 ppbv are still observed in the plume. As seen in Figure 1, much of the land cover outside of Manaus is tropical forest, with some pasture land interspersed. Significant isoprene emissions are expected and indeed measured from the forest (Gu et al., 2017;Guenther et al., 2006). Significant concentrations of methacrolein, methyl vinyl ketone, and isoprene hydroxyhydroperoxides (PTR-MS m/z 71), all first generation oxidation products of isoprene, are also observed and have a more complicated structure (Liu et al., 2016). Concentrations of these products tend to be highest near the edges of the plume and they do not correlate strongly with either isoprene ($r^2 = 0.24$) or with anthropogenic plume markers such as CO ($r^2 = 0.24$). Production of these oxidation products requires OH, the concentration of which depends on $NO_x$, to oxidize isoprene and thus they are expected to have a more complex

relationship with the urban plume and the surrounding biogenic emissions. Taken together, the observations of both isoprene and its oxidation products suggest that photooxidation of BVOCs is enhanced in the plume relative to the background, as would be expected (Martin et al., 2017). Monoterpene concentrations (not shown) on this flight are approximately a factor of ten lower than isoprene concentrations, consistent with previous measurements (Kesselmeier et al., 2000), and are often near

or below the instrument limit of detection (~200 pptv). Monoterpene concentrations correlate with isoprene concentrations during the flight, when they are above the instrument detection limit.

Organic aerosol (OA) mass increases from background values of 0.5 µg/m$^3$ to as high as 3.1 µg/m$^3$ in the Manaus plume and on this day it is correlated relatively well with both ozone (r$^2$= 0.65) and CO (r$^2$=0.76). While OA concentrations do not reach the levels that are typically seen in the outflow of large North American cities, they are significantly larger than background

concentrations (up to 6x), indicating a clear impact of the plume on OA concentrations. Sulfate also clearly increases in the plume, though its peak concentrations occur toward the southern edge of the plume, relative to organics, CO, and ozone, suggesting sulfur emissions may not be co-located with other anthropogenic emissions. However, there are also occasional peaks in the sulfate concentration (for example at 10:47) that coincide with minima in other plume indicators, such as CO and ozone. Nitrate concentrations are significantly lower than sulfate concentrations, but tend to correlate more strongly with

organic concentrations than sulfate concentrations. There are indications that a significant portion of the nitrate mass exists as organic nitrates, consistent with measurements at the T3 site and with aircraft measurements (Allan et al., 2014;de Sá et al., 2018;Schulz et al., 2018). The NO$^+$:NO$_2^+$ ion ratio is 5-7 in the plume, compared to a NO$^+$:NO$_2^+$ ion ratio of 1.2-1.5 measured during the campaign for ammonium nitrate calibration aerosol. NO$^+$:NO$_2^+$ ion ratios significantly above that of ammonium nitrate are indicative of the presence of organic nitrates, though quantifying the organic fraction is challenging (Farmer et al.,

2010). Ammonium concentrations tend to closely mirror sulfate concentrations. The bottom panel of Figure 5 shows the fractional composition of the aerosol, which helps to illustrate the relative differences among the aerosol species. Organic species clearly dominate at all points, with sulfate the next most important species.

At 1000 m altitude, the plume can still be observed in the G-1 data, though the concentrations of plume markers such as CO, ozone, and benzene are lower. Increases in organic concentration are less sharp and defined than observed at 500 m, though

enhancements are still clear. The organic mass fraction is somewhat smaller at 1000 m and sulfate and ammonium mass fractions are larger. Based on radiosonde measurements, the boundary layer grew from 900 to 1200 m between the beginning and end of the flight on this date. Data from the G-1 suggest that the 1000-m flight path was near the top of the boundary layer. Thus, at 1000-m altitude, the data are influenced by both the local conditions and from air that has been transported over longer distances.

Figure 6 shows a PMF analysis of the organic species, for the March 13[th] flight along with independently measured species that might be expected to correlate with the PMF factors. We were able to resolve the organic aerosol into three factors; an OOA factor that is a proxy for SOA, an HOA factor that is a proxy for primary organic particulate emissions, and a IEPOX SOA factor, which forms from the heterogeneous uptake of IEPOX, a product of isoprene oxidation under low-NO$_x$ conditions (Hu et al., 2015;Robinson et al., 2011;Lin et al., 2012;Zhang et al., 2011;Zhang et al., 2005a;de Sá et al., 2017). Details of the

PMF analysis, including mass spectra of the factors, can be found in the appendix and supporting information. As expected, both the OOA and HOA factor mass loadings are significantly higher in the plume relative to the background. The HOA factor correlates strongly with CO (r=0.79 for all data at 500 m) while the OOA factor correlates strongly with ozone (r=0.81 for all data at 500 m), though we will show below that the slope of the correlation changes with plume age. The changing slope has the effect of degrading the HOA/CO and OOA/ozone correlations, which are higher for individual plume legs. The HOA factor is a larger fraction of the organic mass on the legs nearest to Manaus and becomes a smaller fraction on the downwind legs. The OOA mass fraction displays the opposite behavior, increasing downwind as the plume ages. The transformation of HOA to SOA/OOA has been observed previously, both in laboratory experiments and in the field (Sage et al., 2008;Robinson et al., 2007;DeCarlo et al., 2008;Zhang et al., 2007;Ng et al., 2010). Taken together, these observations are consistent with the literature assignment of HOA as a marker of primary organic emissions and OOA as a marker of secondary organic aerosol. Outside of the plume, HOA concentrations approach zero. OOA concentrations on the other hand, remain significant and approach the organic aerosol background concentrations. The IEPOX SOA factor does not appear to change with passage through the plume and does not correlate with sulfate aerosol on this day as may be expected based on the chemical mechanism, which requires heterogeneous uptake of gas-phase IEPOX onto inorganic aerosol to generate IEPOX SOA (Hu et al., 2015;Surratt et al., 2010;Gaston et al., 2014;Nguyen et al., 2014). We note that the IEPOX SOA factor is noisy, is near or below the limit of detection of organic aerosol (0.13 μg/m$^3$) for much of this flight, and is shown largely for the sake of completeness. Because of the low concentrations, it is difficult to draw definitive conclusions about IEPOX SOA formation on this flight. However, given the focus of this flight was on tracking the Manaus plume which contains significant NO$_x$, our findings are consistent with de Sa et al. (2017) who report that NO$_x$ from the Manaus plume suppresses IEPOX SOA formation in the plume and that the IEPOX-SOA factor was less than 5% of the total OA at T$_3$ at the time of G1 overflight.

Figure 7summarizes several relevant plume quantities at 500 m altitude as a function of the approximate distance downwind from Manaus, which serves as a proxy for the photochemical age of the air mass. The ratio of excess organic aerosol to excess CO (Δorg/ΔCO) is a metric that is often used to quantify organic aerosol formation in a source plume as it ages. The utility of the Δorg/ΔCO metric rests mainly the assumptions that CO is conserved on the timescale of the measurements and that urban emissions scale linearly with CO. Using this ratio rather than absolute concentrations of organic aerosol can normalize for dilution due to mixing of the plume with background air. The Δorg/ΔCO values shown in Figure 7 were calculated as the slope of a linear regression line between the AMS-measured organic mass loadings and CO concentrations. All data for each flight leg perpendicular to the wind direction were included. Background values of OA and CO were not subtracted from the data and the regression was not forced through the origin. We acknowledge that calculation of Δorg/ΔCO values can be sensitive to the calculation method and choice of background values for both OA and CO; therefore, we performed calculations using different methods and assuming different background concentrations for both OA and CO and find general agreement between these methods (see SI for more details)..

Two aspects of the Δorg/ΔCO distinguish these measurements from most previous measurements in the literature. First, Δorg/ΔCO is ~31μg/m$^3$ ppmv$^{-1}$, a value that is lower than the value of 70 ± 20 μg/m$^3$ ppmv$^{-1}$ observed in the outflow of cities

in North America (e.g., de Gouw and Jimenez (2009) and refs therein). The $\Delta org/\Delta CO$ values observed here are also significantly smaller than the values of $77 - 157 \ \mu g/m^3 \ ppmv^{-1}$ we previously observed when fresh urban anthropogenic emissions evolved in the presence of strong biogenic emissions, as is the case in the present data set (Shilling et al., 2013;Setyan et al., 2012). We measure a $\Delta HOA/\Delta CO$ value of $17.6 \ \mu g/m^3 \ ppmv^{-1}$ near Manaus (Figure 7) which should represent fresh emissions. This value is at the upper end of what is reported in the literature for North American cities, indicating that lower primary particulate emissions from Manaus are not responsible for the lower $\Delta org/\Delta CO$ values (DeCarlo et al., 2010;de Gouw and Jimenez, 2009).

The second new aspect of these observations is that we do not observe a corresponding increase in $\Delta org/\Delta CO$ as the plume photochemically ages, contrary to our expectations. We note that, as described above, organic aerosol concentrations are significantly higher in the plume than in the background and the constant $\Delta org/\Delta CO$ values do not indicate that no OA formation occurs in the plume. Rather, the $\Delta org/\Delta CO$ analysis that follows focuses on understanding the aging of the Manaus plume. Based on literature observations (de Gouw and Jimenez, 2009;Kleinman et al., 2008;DeCarlo et al., 2010;Takegawa et al., 2006;Sullivan et al., 2006;Freney et al., 2014), we hypothesized that $\Delta org/\Delta CO$ would rapidly increase in the Manaus plume due to OA formation from sources not associated with urban CO emissions (Setyan et al., 2012;Shilling et al., 2013). Specifically, we expected that enhanced OH concentrations in the plume would lead to rapid oxidation of BVOCs emitted from the surrounding forest, which would produce OA with little concomitant production of CO. As seen in Figure 7, the data did not support this hypothesis. As described in the Supporting Information, we performed the $\Delta org/\Delta CO$ calculation using several methods and assuming different background CO and OA values and did not find significant increases in $\Delta org/\Delta CO$ in any of these calculations. Furthermore, though we focus on $\Delta org$ values based on the AMS data, we also calculated $\Delta volume/\Delta CO$ using aerosol size distribution data from two independent instruments, the Ultra-High Sensitivity Aerosol Spectrometer (UHSAS) and the Fast Integrated Mobility Spectrometer (FIMS) that were also on board the G-1. Neither of these calculations indicates an increase in $\Delta volume/\Delta CO$ with plume age.

Previous studies have observed increases in $\Delta org/\Delta CO$ with plume age, with the largest increases occurring for the first ~1 day of aging and changes in $\Delta org/\Delta CO$ gradually levelling off beyond approximately 1-2 days (DeCarlo et al., 2010;de Gouw and Jimenez, 2009;Kleinman et al., 2008;Freney et al., 2014). Based on the mean wind speeds observed along the flight track (7.3 m s$^{-1}$) and the transport distance (up to 100 km), we estimate that the plume was 4-5 hours old at the farthest leg and freshly emitted over the city. Unfortunately, photochemical clocks could not be used to more precisely calculate the photochemical age of the plume (de Gouw et al., 2005;Kleinman et al., 2003;Parrish et al., 1992). NO$_y$ measurements were not available on this flight. Benzene and toluene concentrations were low and noisy, particularly at increasingly downwind distances from Manaus.. Though our observations are limited to shorter aging timescales (4-5 hours) than many literature studies (1-2 days), the literature studies report measurable changes in $\Delta org/\Delta CO$ at short aging timescales (Kleinman et al., 2008;DeCarlo et al., 2010). Furthermore, we clearly observe other indicators of photochemical aging in the plume. Ozone concentrations are 30 – 50 ppbv in the plume, compared to background levels of 10-15 ppbv, indicating active photochemistry. Concentrations of isoprene are lower in the plume than the background values observed outside of it. Concentrations of

isoprene inside the plume do not show a monotonic dependence on plume age, The concentration of isoprene oxidation products measured in the plume by the PTR-MS at m/z 71 monotonically increases with plume aging, from 0.96 ppbv directly over the city to 1.27 ppbv at the farthest leg. The average toluene:benzene ratio in the plume also monotonically decreases with plume aging as would be expected, though attempts to convert these ratios into a photochemical age resulted in unrealistically high estimates of the plume age, likely due to noise in the data as mentioned above (de Gouw et al., 2005). The mean particle oxidation state ($\overline{OS}_c$) of plume OA increases from -0.6 to -0.44 as it ages. It is expected that photochemical aging would produce progressively more oxygenated species downwind of Manaus that subsequently partition to the aerosol phase, increasing the mean carbon oxidation state. Other studies have observed a similar phenomenon as particles were transported downwind of urban centers (DeCarlo et al., 2010). The particles size distributions measured by both the FIMS and the UHSAS also indicate that particles size increases downwind of Manaus. As discussed in the previous section, we also observe that the mass fraction of HOA decreases and simultaneously the mass fraction of OOA increases as the plume ages. As seen in Figure 7, $\Delta$HOA/$\Delta$CO is 17.6 µg/m$^3$ ppmv$^{-1}$ on the leg nearest to the city and decreases to 10.6 µg/m$^3$ ppmv$^{-1}$. At the same time, $\Delta$OOA/$\Delta$CO increases from 9.2 µg/m$^3$ ppmv$^{-1}$ to 23.1 µg/m$^3$ ppmv$^{-1}$. Thus, some fraction of HOA appears to be lost, either through volatilization, deposition, or a combination of both, with the lost HOA mass balanced by an increase in OOA mass. All of these factors indicate that active photochemistry is occurring and is transforming VOC's into SOA and aging the particles. Thus, we would expect to see significant increases in $\Delta$org/$\Delta$CO for these aging times, if similar observations of outflow from North American cities were representative of the Manaus plume.

The sum of the HOA and OOA factors explains >95% of the total OA mass within the plume, as the IEPOX SOA mass is small. Though calculating the $\Delta$HOA/$\Delta$CO and $\Delta$OOA/$\Delta$CO ratios introduces some noise (particularly $\Delta$OOA/$\Delta$CO close to Manaus), the sum of these ratios is approximately constant with aging and equal to the $\Delta$org/$\Delta$CO values. Thus, the total OA mass is conserved with loss and volatilization of HOA balanced out by formation of OOA. This process appears to explain the lack of an increase in $\Delta$org/$\Delta$CO with aging. It is difficult to determine whether the OOA carbon mass is anthropogenic in origin, biogenic, or (most likely) a combination of both. Volatilized HOA and co-emitted anthropogenic VOCs would undergo oxidation in the plume and the products may condense as OOA (Sage et al., 2008;Robinson et al., 2007). In addition, we see evidence that biogenic VOCs are oxidized in the plume, likely more rapidly than in the background due to enhanced in-plume OH concentrations, and these oxidation products may also condense as OOA. Either mechanism or a combination of both would explain the observed increase in the carbon oxidation state. It is surprising that the volatilized HOA mass is compensated by oxidation and re-condensation of SOA for these relatively fresh particles.

Thus far, we have focused on a detailed analysis of the March 13, 2014 plume because there is a clear contrast between the background and plume data through the portion of the flight in the boundary layer, there is little interference from biomass burning and cloud processing, the flight pattern was extended far enough downwind to investigate a significant aging time, and the data from all key instruments are available and complete. In an effort to understand whether the $\Delta$org/$\Delta$CO observations on March 13 are representative of plume aging in the wet season, we have screened the rest of the wet season flight data and, based on the criteria mentioned above, we have identified two additional flights, March 11 and March 16, 2014, for analysis.

The $\Delta$org/$\Delta$CO values for these data as a function of distance from Manaus are shown in Figure 8. On March 11, $\Delta$org/$\Delta$CO values average 29.9 $\mu$g/m$^3$ ppmv$^{-1}$, show little change with distance from Manaus, and show similar variability as the March 13 flight data. The mean wind speed for the analyzed portion of the flight was 5.3 m/s translating to an approximate maximum plume age of 5-5.5 hours. On March 16, $\Delta$org/$\Delta$CO values average 42.0 $\mu$g/m$^3$ ppmv$^{-1}$, show little significant increase with distance from Manaus, and exhibit a larger range of values than the other two datasets. The mean wind speed for the analyzed portion of the March 16 flight was 4.6 m/s translating to an approximate maximum plume age of 6-6.5 hours. We estimate that the error in the $\Delta$org/$\Delta$CO measurement is approximately 10 $\mu$g/m$^3$ ppmv$^{-1}$, thus the difference in average $\Delta$org/$\Delta$CO values between the March 16 and the March 11 and 13 datasets are at the edge of what we would consider significant. The average $\Delta$org/$\Delta$CO is 34.3 $\mu$g/m$^3$ ppmv$^{-1}$ for all flights and collectively represent the first 4 – 6.5 hours of aging. In addition to the aircraft data, Cirino et al. (2018) report that the median $\Delta$org/$\Delta$CO values for the plume as measured at the T2 and T3 sites are nearly identical to one another in the wet season and similar to measurements from the G-1(Cirino et al.). Thus, the data from both G-1 and the ground sites suggests that $\Delta$org/$\Delta$CO values of 30-40 $\mu$g/m$^3$ ppmv$^{-1}$ that change little in the first 4-6.5 hours of plume aging are representative of the Manaus plume behaviour in the wet season. As previously discussed, most studies examining outflow of North American cities have shown changes in $\Delta$org/$\Delta$CO that are largest at the shortest aging times and we are unaware of reports of constant $\Delta$org/$\Delta$CO in an aging urban plume. Both the identity and the distribution of Manaus's emissions may very well be distinct from that of a typical North American city. In addition, background concentrations of anthropogenic pollutants are lower and biogenic VOC emissions are higher in the unperturbed rain forest surrounding Manaus relative to most North American cities. Finally, background OA concentrations near Manaus are lower in the wet season than in most other continental regions and SOA yield is known to increase with OA mass loading (Odum et al., 1996). All of these factors may contribute of the difference in the $\Delta$org/$\Delta$CO observations.

A similar phenomenon, clear chemical aging with little change of $\Delta$org/$\Delta$CO, has been observed in biomass burning plumes by several researchers (Jolleys et al., 2012;Cubison et al., 2011;Hecobian et al., 2011;Akagi et al., 2012;Forrister et al., 2015). Clearly, the mix of organic compounds emitted from a forest fire and from the Manaus urban region will be significantly different and this is not to imply that the detailed chemical mechanisms are the same. However, the similarity of the observations is suggestive that a similar process could occur in both types of plumes.

### 3.3 Sources of Sulfate in the Manaus Region

Sulfate was the second largest contributor to the non-refractory aerosol mass during the GoAmazon 2014/5 campaign (accounting for 13% of the PM$_1$ mass in both the wet and dry season), so it is also instructive to examine the source of sulfate in the area. In 2014 Manaus generated much of its power from fuel oil and diesel, which are known to emit SO$_2$ that eventually oxidizes to form condensable sulfate (Medeiros et al., 2017). However, several observations suggest that the major source of sulfate in the plume may not come from local SO$_2$ emissions. First, on many flights, particulate sulfate levels clearly increases in the plume (e.g., Figure 5), including on the legs closest to Manaus and thus very near the source. The lifetime of SO$_2$ with respect to homogeneous oxidation by OH is expected to be about one week and only small fraction (~5% on the farthest leg)

of $SO_2$ emissions would have time to oxidize to $H_2SO_4$ in the gas phase. In addition, there is no indication of dramatically enhanced sulfate loading downwind of Manaus on the March 13 flight when $\Delta SO_4/\Delta CO$ ratio averages 2.6 µg/m$^3$ ppmv$^{-1}$ and does not increase with plume age. On the March 13 flight, there was no precipitation and relatively few clouds along the flight track so heterogeneous oxidation of sulfate in cloud water cannot explain the sulfate enhancement seen in the plume.

We can possibly attribute these observations to two mechanisms. First, it is possible that natural sulfur emissions originating downwind of Manaus undergo oxidation by enhanced concentrations of OH in the Manaus plume and subsequently condense (Andreae et al., 1990;Chen et al., 2009). Andrea et al. (1990) estimate that in-basin sulfur emissions are responsible for approximately 0.05 µg/m$^3$ of the total sulfate loading. Measurements during the AMAZE-08 campaign also suggested natural in-basin emissions and transport of natural out of basin emission were a significant source of particulate sulfate (Chen et al.,

2009). A 0.05 µg/m$^3$ sulfate loading is roughly consistent with our measured sulfate concentrations outside of the plume and would account for 15 – 30% of the sulfate in the plume on the March 13 flight. Higher OH concentrations in the plume would speed oxidation of natural sulfur compounds and further increase the fraction of in-plume sulfate originating from natural emissions. Second, power plant plumes may directly emit sulfuric acid, which then may condenses on pre-existing particles. A previous aircraft study of the Manaus outflow observed high concentrations of particles smaller than 40 nm in power plant

plumes, which they attribute to sulfuric acid (Kuhn et al., 2010). Though particles in this size range are unlikely to be detected by the AMS, condensation of sulfuric acid on larger, pre-existing particles would also contribute to the in-plume sulfate observations.

## 4 Summary and Implications

In summary, we report on measurements of the aerosol chemical composition, sources, and evolution of the urban plume from

35 research flights conducted in both the wet and dry season in the vicinity of Manaus, Brazil. Median mass loading of organics, sulfate, nitrate, and ammonium were 0.85, 0.19, 0.02, and 0.06 µg/m$^3$ in the wet season and 4.29, 0.77, 0.05, and 0.25 µg/m$^3$ in the dry season. Despite the significant difference in mass loadings, the average fractional composition of the aerosol did not change significantly between seasons. Organics dominate the aerosol chemical composition in both seasons, comprising 80% of the total non-refractory aerosol mass. The OA was significantly more oxidized in the dry season with a $\bar{O}S_c$ of -0.02 in the

dry season and -0.45 in the wet season.

The flight on March 13, 2014 was a golden day to study the evolution of the Manaus plume as it advected to the surrounding Amazon tropical forest. The organic and sulfate aerosol concentrations were both significantly enhanced in the plume along with CO, ozone, and benzene. The spatial distribution of the sulfate aerosol was shifted toward the southern edge of the plume, relative to organics, CO, and benzene. As the plume is transported downwind of Manaus, we observe a change in the relative

fraction of HOA and OOA mass. $\Delta HOA/\Delta CO$ values decreased from 17.6 µg/m$^3$ ppmv$^{-1}$ over Manaus to 10.6 µg/m$^3$ ppmv$^{-1}$ 95 km downwind of Manaus, indicating a substantial loss of HOA mass. This loss of HOA mass was balanced out by OOA formation, with $\Delta OOA/\Delta CO$ values increasing from 9.2 to 23.1 µg/m$^3$ ppmv$^{-1}$ during the 4-5 hour aging timescale.

Concomitantly, the mean carbon oxidation state of the OA increased from -0.6 to -0.44. Because the loss of HOA mass was balanced out by addition of OOA mass, net changes in Δorg/ΔCO with plume age were not observed and averaged 31 μg/m$^3$ ppmv$^{-1}$. Analysis of data from two additional fights during the wet season also found no net changes in Δorg/ΔCO with plume age. The average Δorg/ΔCO for all the anayzed G-1 data was 34 μg/m$^3$ ppmv$^{-1}$ and collectively represent the first 4- 6.5 hours

of plume aging. The observation of constant Δorg/ΔCO with aging was in contrast to our hypothesis that Δorg/ΔCO would increase rapidly due to plume-enhanced oxidation of BVOCs, emitted by the surrounding tropical forest and not associated with CO emissions, and subsequent conversion to OA mass. Our Δorg/ΔCO observations are also in contrast to literature observations of the outflow of several different North American urban centers, which have shown increases in Δorg/ΔCO for the first 1-2 days of plume aging (Kleinman et al., 2008;de Gouw and Jimenez, 2009;DeCarlo et al., 2010;Freney et al.,

2014;Sullivan et al., 2006).

The dataset generated from the G-1 measurements during GoAmazon 2014/5 provide a set of observations for understanding aerosol chemistry in the Amazon region. Our preliminary analysis shows that outflow from the Manaus plume generates less OA in the wet season when compared to outflow of many North American cities. The differences are likely due to a combination of factor including differences in emissions from both Manaus and the surrounding tropical forest, lower levels

of background anthropogenic pollution in the Amazon, and lower background OA concentrations into which semi-volatile organics can partition. These results have implications for modelling efforts and for understanding how urban pollution impacts the surrounding pristine Amazon.

**Appendix**

**PMF Analysis of AMS data**

Positive Matrix Factorization (PMF) analysis was performed on the high-resolution organic aerosol mass spectra and error matrix that were measured using the HR-ToF-AMS on the G-1. The organic aerosol mass spectra (m/z < 122) and error matrix were initially prepared using the high resolution AMS analysis package (Squirrel v1.55H, PIKA v1.44H) as described in the literature (Jimenez et al., 2003;Allan et al., 2004;DeCarlo et al., 2006;Aiken et al., 2007;Aiken et al., 2008). As described in the main text, the AMS organic mass loading data were normalized to the laboratory calibration conditions of 23 °C 1013 hPa.

Error was estimated from the ion counting statistics (Allan et al., 2003) and a minimum error was assigned to each signal (Ulbrich et al., 2009). Ion with signal below zero (due to noise) were removed from the matrix. In addition, ions with an average signal to noise (S/N) less than 0.75 were removed and ions with average S/N between 0.75 and 2 were downweighted by a factor of two (Ulbrich et al., 2009). Ions whose signals are fixed based on the measured $CO_2$ signal in the ratios described by Aiken et al., (2008) ($CO^+$, $H_2O^+$, $HO^+$, $O^+$) are also downweighted by a factor of two (Ulbrich et al., 2009). The PFM2

algorithm version 4.2 (Paatero, 1997;Paatero and Tapper, 1994) was used to analyze the matrix and results were visualized and evaluated using the PMF Evaluation Tool (v 2.06) described by Ulbrich et al. (2009). PMF analysis of this data set was

challenging, due the combination of low concentrations of organic aerosol, fast sampling required for aircraft operations, and the often small changes in aerosol concentrations across the flight domain. PMF analysis on a data from a single flight often (though not always) resulted in a failure to resolve factors that are described in the literature (e.g., HOA, IEPOX SOA). Instead, we often observes split factors that are largely dominated by a single high S/N ion ($CHO^+$, $CO_2^+$, $C_2H_3O^+$). For example, the

IEPOX SOA factor was not resolved when the March 13 flight was analyzed on its own, largely due to the small mass loading of this factor (Figure 6). Therefore, data from all flights during the wet season was combined into a single matrix and analyzed together, which also forces a common solution for the entire dataset.

We chose a five factor solution to the combined matrix based on number of criteria, including; the residuals and fit quality, the correlation of chosen factors with independently measured species (e.g., CO, ozone), by examining the factor mass spectra for

realism (are the spectra representative of a physical compound?), and by comparing the factor spectra to literature data. Table A1 summarizes the residuals as a function of the number of factors and a short summary of the results. First, as seen in Table A1, the residuals continue to significantly decrease in moving from a 2 to 5 factor solution. For example, $Q/Q_{exp}$ decreases from 0.28 to 0.24 and the fraction of the residual aerosol mass decreases from 4% to 0.1% in moving from the two to the five factor solution. Figure S2 shows the residuals and scaled residuals for each time point in the matrix for the 5 factor solution,

indicating that all the data are well fit. Solutions with three or fewer factors failed to separate the HOA and the IEPOX SOA factors, which are well described in the literature and were reasonably expected to be present in the data. One of the OOA-like spectra in the three factor solution also appeared to have signs of both the HOA and IEPOX SOA factor in the spectra, as evidenced by significant signal of the key marker ions $C_4H_9^+$ (m/z 57) for HOA and $C_5H_6O^+$ (m/z 82) for IEPOX SOA. In the four factor solution, an HOA factor was resolved along with three OOA-like spectra, though the OOA spectra show evidence

of factor splitting (discussed below). The five factor solution was able to resolve an HOA, IEPOX-SOA and three OOA-like factors.

Figure S3 and S4 show the mass spectra and time series for the five factor solution. Examination of the mass spectra (Figure S4) suggests that the OOA factor has split. For example, the mass spectrum of Factor 3 is dominated by the $CHO^+$ ion with little signal through the rest of the spectra. The spectrum of Factor 1 is dominate by $CO_2^+$ and other ions that are based on the

$CO_2^+$ signal ($CO^+$, $H_2O^+$, $HO^+$, $O^+$) in prescribed ratios (Aiken et al., 2008). These factors are unlikely to represent some real aerosol component, as no known OA produces such simple spectra. In addition, the time series of Factors 1 and 2 are highly correlated in time (see Figure S5). In summary, two factors are highly correlated in time (Factors 1 and 2) and a third factor is dominated by a single ion. Thus, we recombine Factors 1, 2, and 3 into a single factor, which we label OOA. The spectra of the resultant three factors (after re-combining) are shown in Figure S6 and the recombined factor is presented in the main text.

The mathematical reason for the factor splitting is clear; aerosol loading is low, the AMS is sampling at a relatively fast rate (13 s averaging time), and the AMS spectra are dominated by a relatively small number of high S/N ions. In separate analyzes, we downweighted the $CHO^+$, $C_2H_3O^+$, and $CO_2^+$ ion signals by factors of 2-10 in an attempt to minimize the splitting of the OOA factor, but were unsuccessful.

Comparison of the factor mass spectra to literature PMF spectra and correlation of the factor time profiles with independent measurements provides further support for the 5-factor solution. The combined OOA factor compares well with the spectra presented in the literature for OOA (e.g., (Zhang et al., 2007;Zhang et al., 2011;Zhang et al., 2005b;Ng et al., 2010). As shown in Figure 6, the OOA factor correlates well with ozone, a known secondary species that also forms through photochemical reactions. Similarly, the spectrum of Factor 4 compares well with the mass spectrum of HOA widely reported in the literature (e.g., (Zhang et al., 2007;Zhang et al., 2011;Zhang et al., 2005b;Ng et al., 2010). As discussed in the text and shown in Figure 6, the HOA factor correlates well with CO, a known tracer of anthropogenic combustion processes. We label Factor 5 as the IEPOX SOA factor that is widely reported in the literature and believed to represent SOA formed from the heterogeneous uptake of isoprene epoxydiols onto pre-existing aerosol (Hu et al., 2015;Robinson et al., 2011;Lin et al., 2012). Our assignment is based on comparison with the factor mass spectral profile with those in the literature e.g., (Hu et al., 2015). 5 Acknowledgements

The authors thank the G-1 flight and ground crews for supporting the GoAmazon 2014/5 mission. Funding for data collection onboard the G-1 aircraft and at the ground sites was provided by the Atmospheric Radiation Measurement (ARM) Climate Research Facility, a U.S. Department of Energy (DOE) Office of Science user facility sponsored by the Office of Biological and Environmental Research (OBER). Data analysis and research was supported by the U.S. DOE's Atmospheric System Research Program under Contract DE-AC06-76RLO 1830 at PNNL. PNNL is operated for the U.S. DOE by Battelle Memorial Institute. We acknowledge the support from the Central Office of the Large Scale Biosphere Atmosphere Experiment in Amazonia (LBA), the Instituto Nacional de Pesquisas da Amazonia (INPA), and the Instituto Nacional de Pesquisas Espaciais (INPE). P. Artaxo acknowledges Fundação de Amparo à Pesquisa do Estado de São Paulo FAPESP grants 2013/05014-0, and 2017/17047-0. The work was conducted under licenses 001262/2012-2 and 001030/2012-4 of the Brazilian National Council for Scientific and Technological Development (CNPq).

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

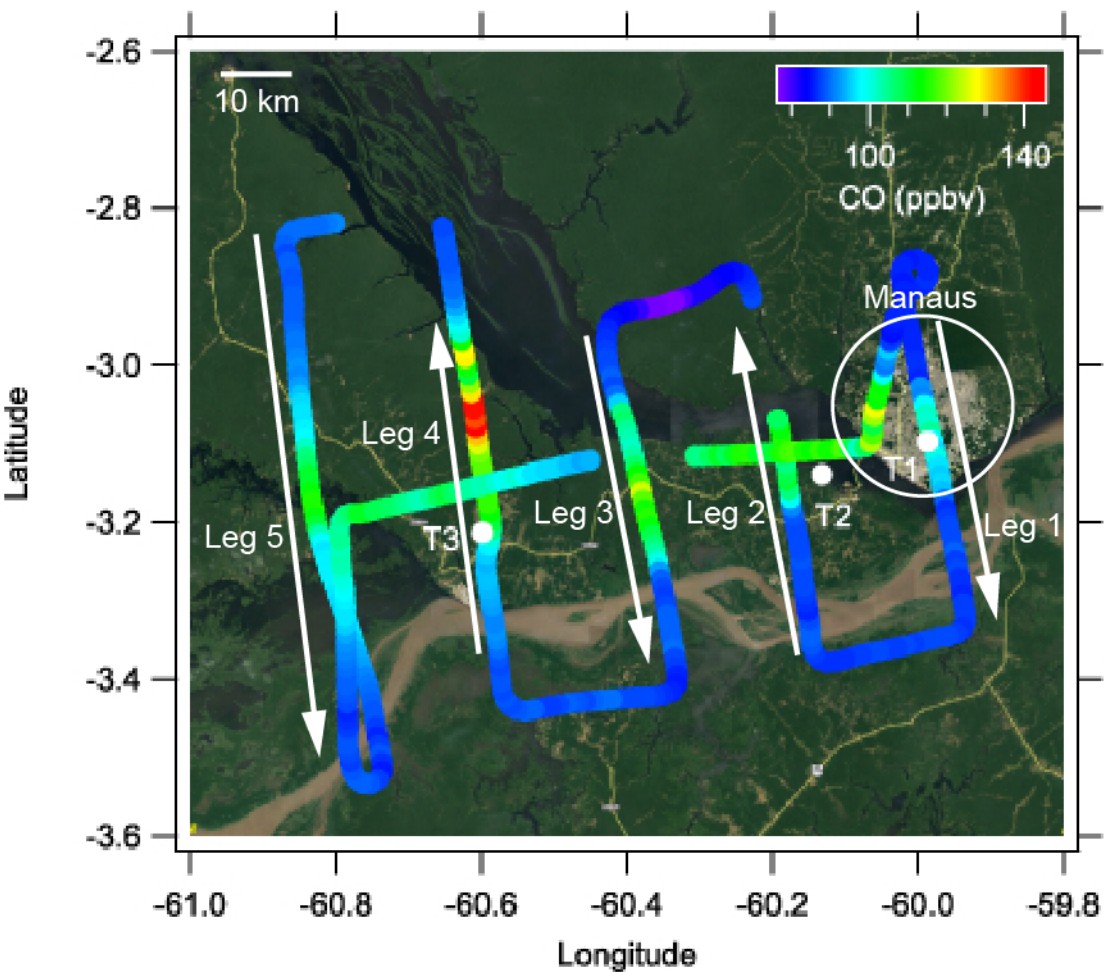

Figure 1. Flight path for the March 13, 2014 flight colored by the CO concentration. Arrows indicate the direction of the flight. The first pass through the pattern was conducted at 500 m then repreated at 1000 m. The G-1 took off at 10:12 (local time) and landed at 13:21 for a total flight duration of three hours and nine minutes. The dots show the location of ground measurement sites. The color scale shows the CO concentration in ppm units. The underlying image is from Google Earth.

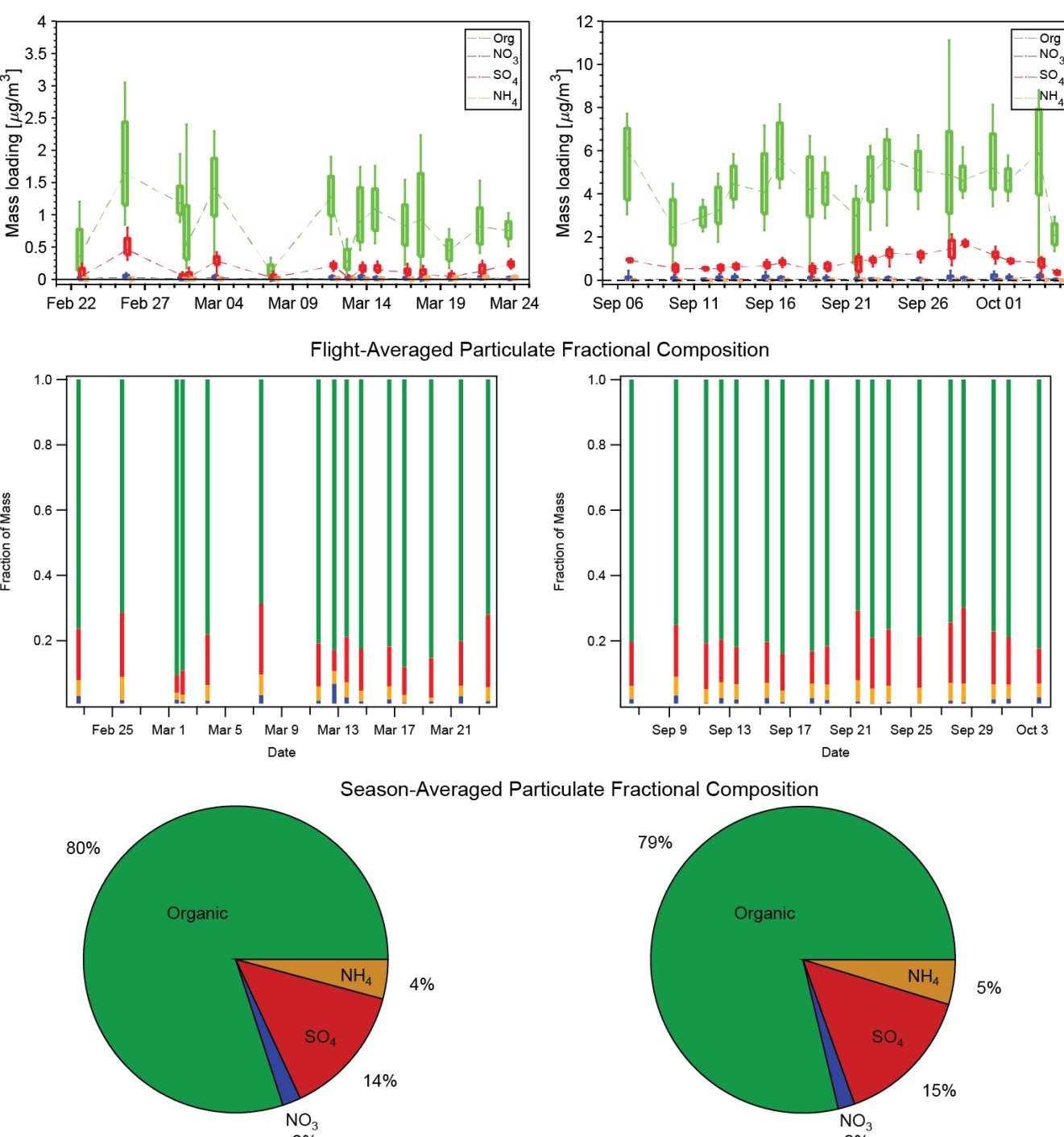

GoAmazon 2014/5 Wet Season      GoAmazon 2014/5 Dry Season

Flight-Averaged Speciated Particulate Mass Loading

Flight-Averaged Particulate Fractional Composition

Season-Averaged Particulate Fractional Composition

**Figure 2: Box and whisker representations of the G-1 AMS data for both the wet and dry seasons (top panel), average particle chemical composition for each flight (middle panel), and average aerosol chemical composition for all flights (bottom). Boxes represent the quartiles, with whiskers extending to 10% and 90%. Lines between the boxes connect the median and are drawn to guide the eye. Note the scale change between the wet and dry season box and whisker plots. AMS data are normalized to 23 °C and 1013 hPa.**

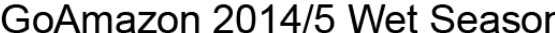

**Figure 3. Box and whisker representations of the G-1 PTR-MS data for both the wet and dry seasons. Boxes represent the quartiles, with whiskers extending to 10% and 90%. Note the scale change between the top and bottom panels.**

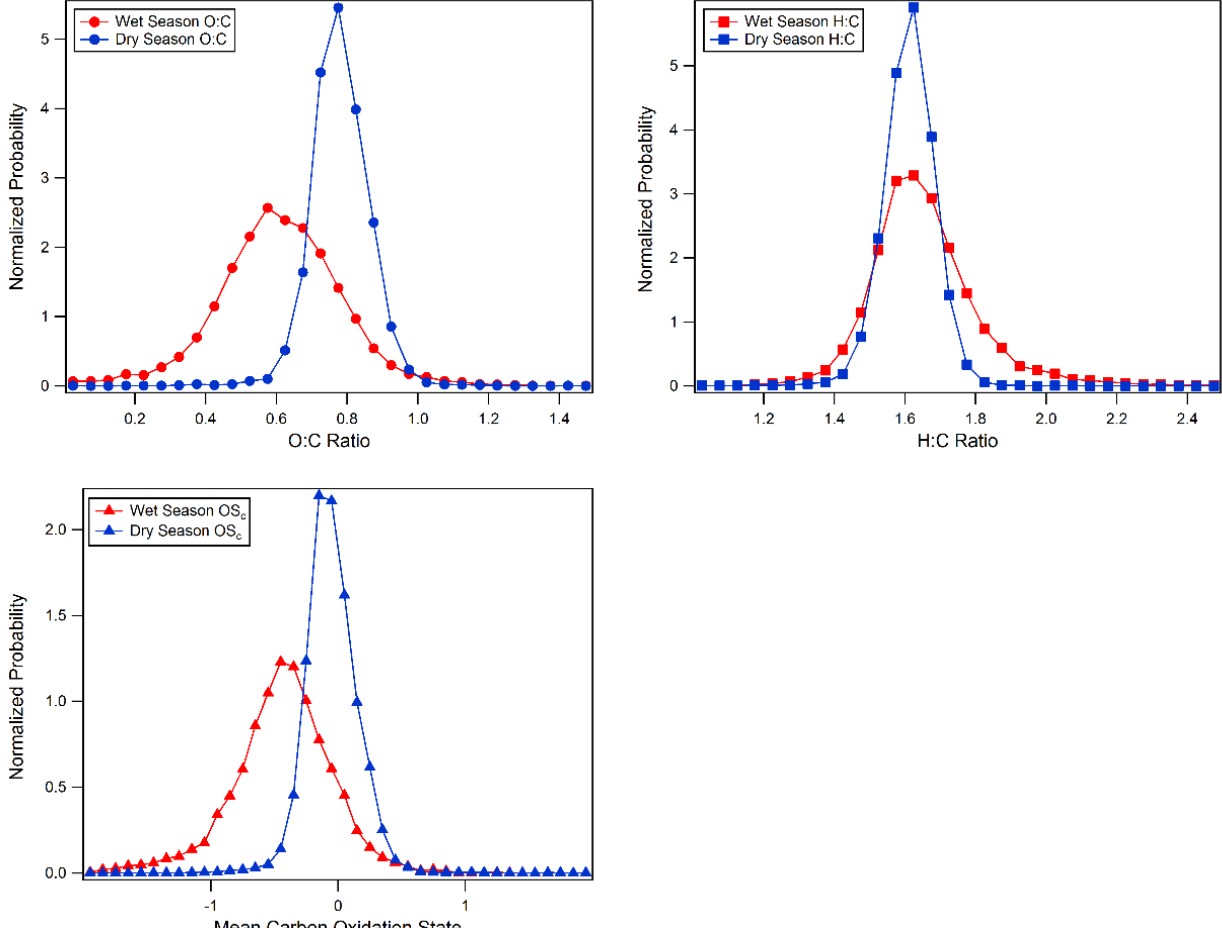

**Figure 4. Normalized occurrence probability calculated from G-1 HR-AMS analysis of the organic aerosol during the wet (red) and dry (blue) seasons. O:C and H:C were binned into 0.05 unit wide bins and $\overline{OS}_c$ data were binned in 0.1 unit wide bins. O:C and H:C ratios were calculated using the updated methodology in Canagaratna et al. 2015. Mean carbon oxidation state is calculated according to Kroll et al. 2011. The same data were analyzed for this figure and for Figure 1 and as such cover a range of altitudes and to a lesser extent time of day.**

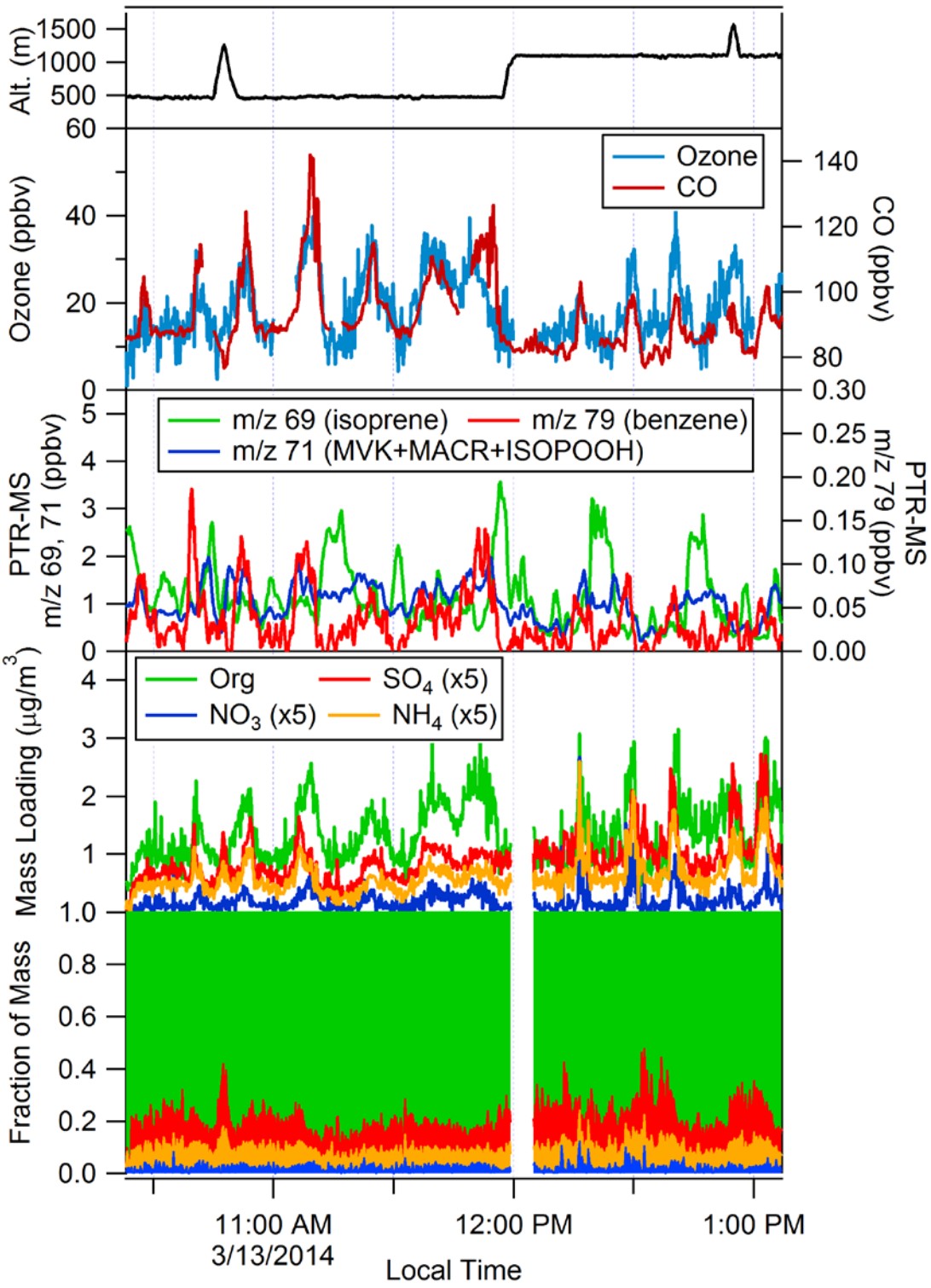

**Figure 5. Time traces of relevant quantities on March 13th, 2014 flight.. Note the mass of inorganic species (SO₄, NO₃, NH₄) has been multiplied by a factor of five to improve the figure clarity. PTR-MS signal at m/z 69 corresponds to isoprene, m/z 71 corresponds to the sum of methylvinyl ketone (MVK), methacrolein (MACR), and isoprene hydroxyhydroperoxide (ISOPOOH), all oxidation products of isoprene, and m/z 79 corresponds to benzene. AMS data are normalized to 23 °C and 1013 hPa.**

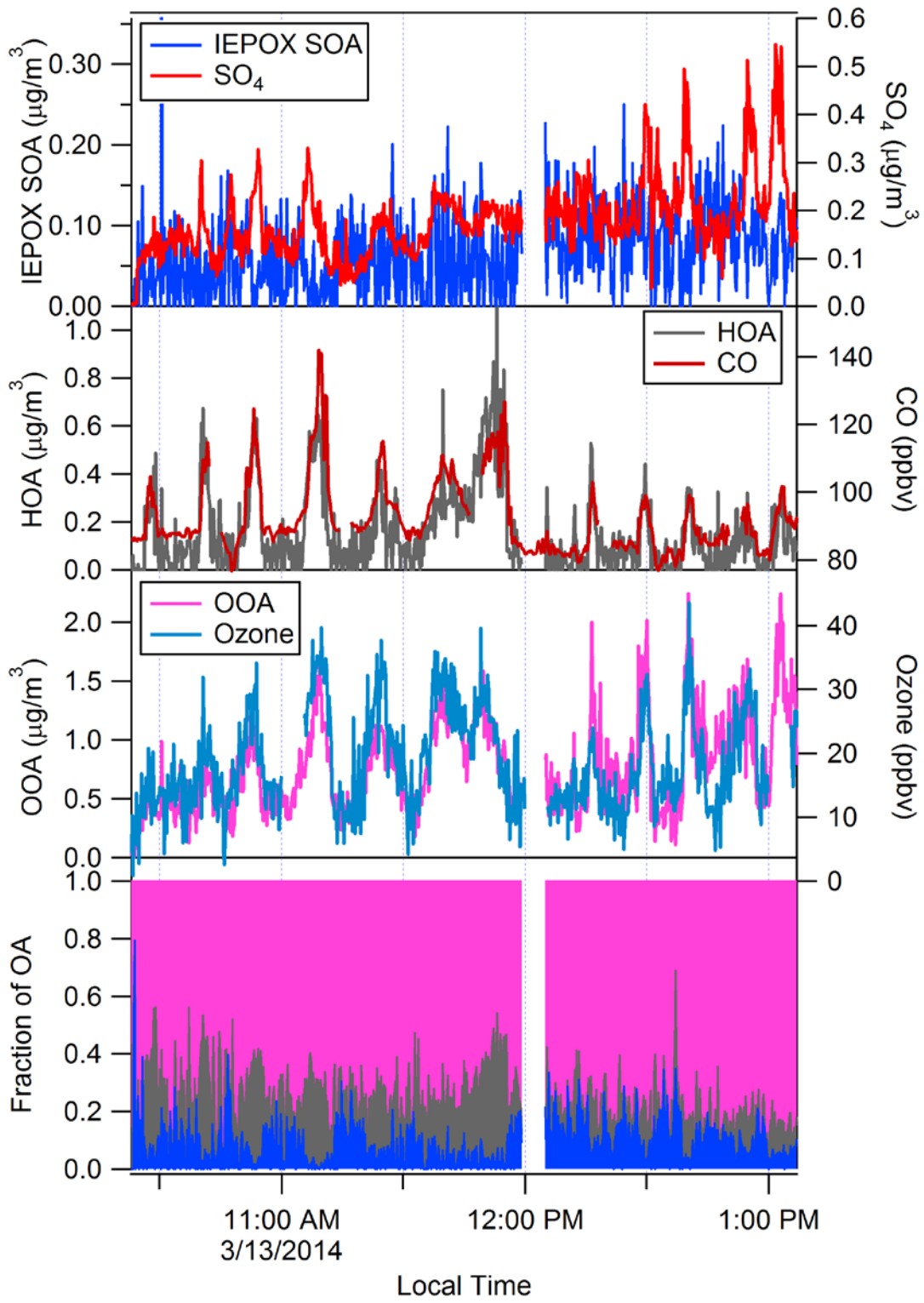

**Figure 6. PMF analysis of the organic aerosol on the March 13th 2014 flight. AMS data are normalized to 23 °C and 1013 hPa.**

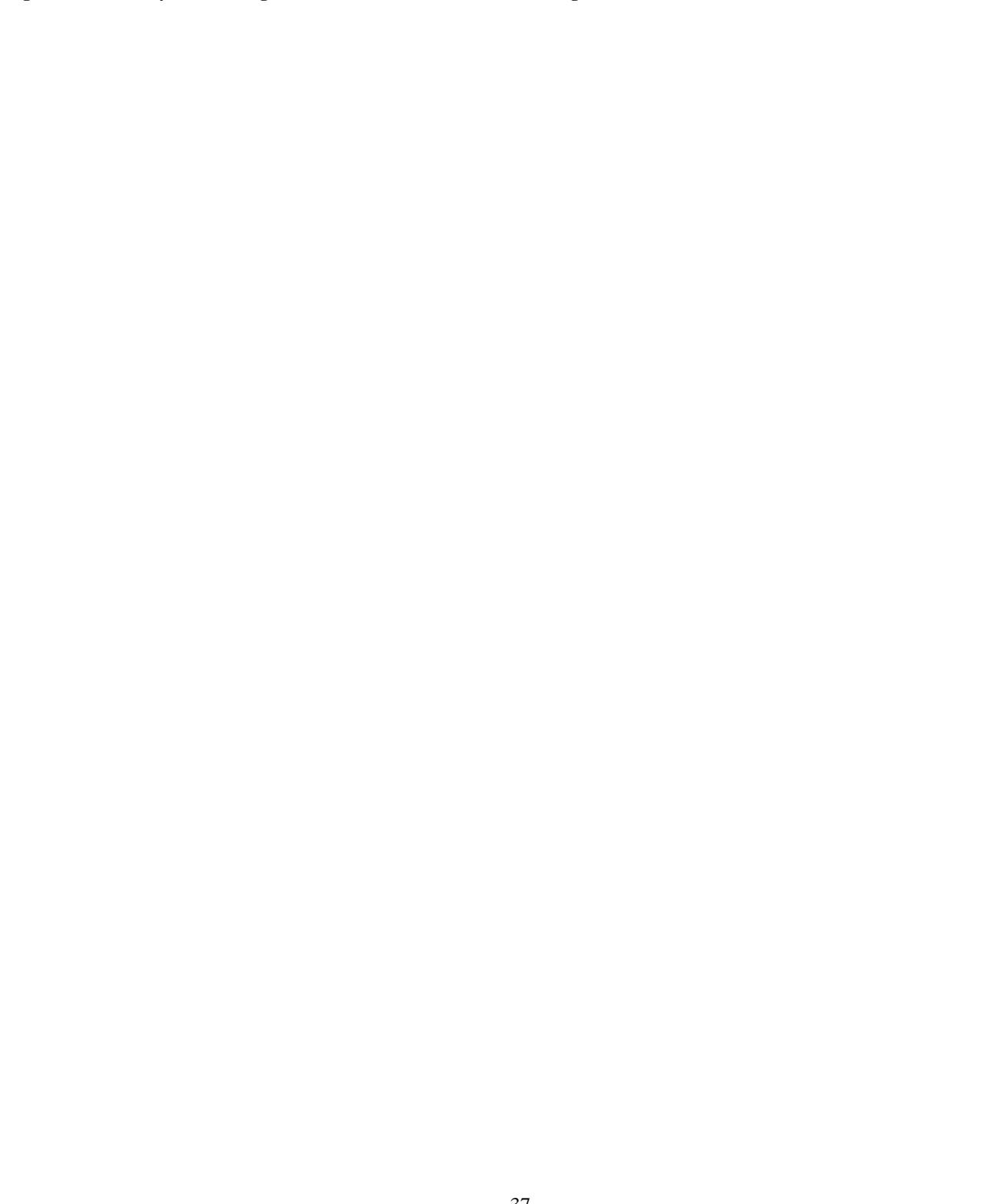

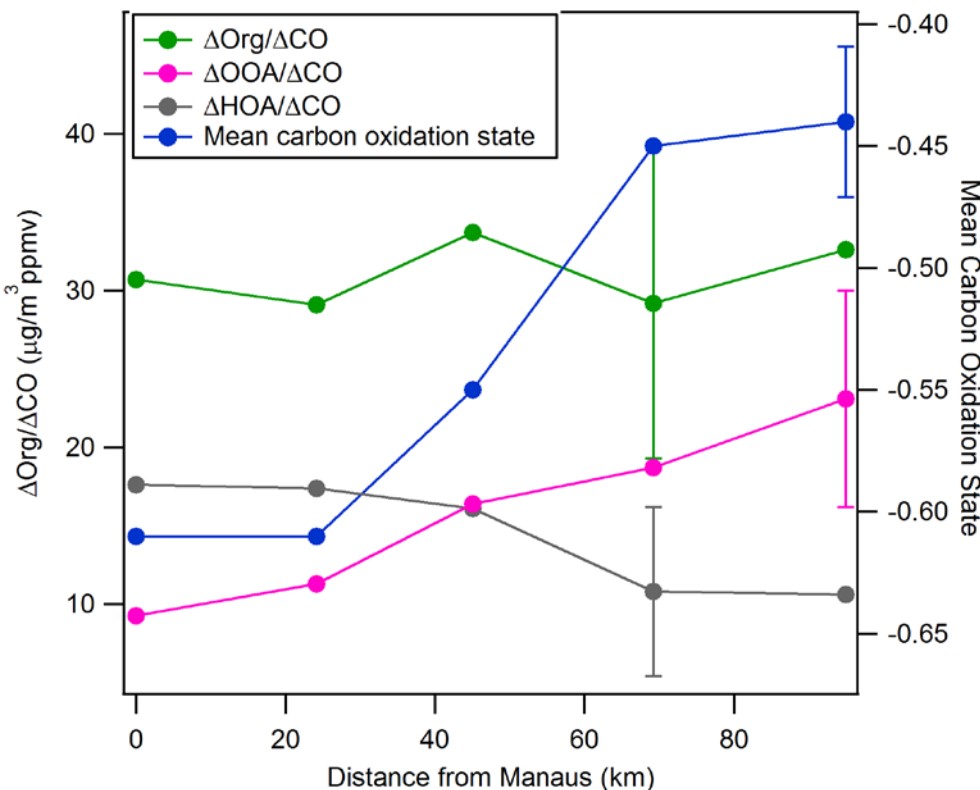

Figure 7. Key metrics describing the evolution of the Manaus urban plume on March 13. Each data point represents the average values for one pass through the plume. A representative error bar is shown for each set of measurements. Mean wind speeds were 7.3 m/s on this flight, thus data capture approximately the first 4-5 hours of the plume aging. Calculations of ΔOrg/ΔCO values use method one with details on the calculations and methods provided in the SI. Spatial locations of each leg are labelled in Figure 1.

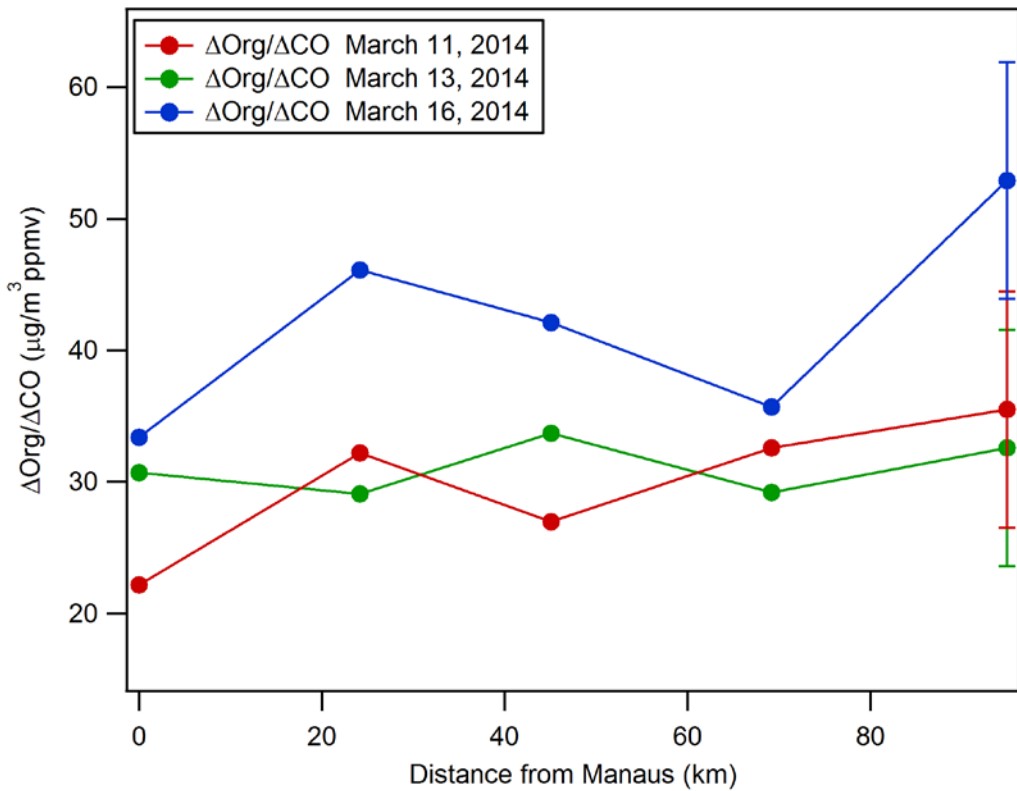

**Figure 8** Δorg/ΔCO measurements in the Manaus plume as a function of distance from Manaus for March 11, March 13, and March 16, 2014. A representative error bar is shown for each set of measurements. Mean wind speeds were 5.8 m/s, 7.3 m/s, and 4.6 m/s on March 11, 13, and 16h, respectively. Collectively, the data capture approximately the first 4-6.5 hours of the plume aging.

| Flight Date[a] | Takeoff [b] | Landing [b] | Altitude(s) of Level Flight Legs (m)[c] | T (°C)[d] | RH (% water)[d] | Wind Speed (m/s)[d] | Wind Heading[d] |
|---|---|---|---|---|---|---|---|
| **Wet Season** | | | | | | | |
| 2/22/2014 | 10:37:11 | 13:25:28 | 600, 1600 | 25 | 80 | 5 | ENE |
| 2/25/2014 | 12:30:31 | 14:43:04 | 600, 1300 | 26 | 80 | 8 | ENE |
| 3/1/2014 a | 9:33:09 | 11:31:31 | 600 | 25 | 80 | 6 | NE |
| 3/1/2014 b | 13:13:59 | 14:49:34 | 1100, 4500, 6400 | 24 | 80 | 4 | NE |
| 3/3/2014 | 13:46:34 | 15:11:57 | 600 | 26 | 90 | N/A | N/A |
| 3/7/2014 | 9:08:11 | 11:39:18 | 600, 1300, 1900, 3200, 3800, 4500, 5800 | 26 | 65 | 10 | NNE |
| 3/11/2014 | 10:39:59 | 13:51:10 | 600, 1500 | 25 | 85 | 7 | ENE |
| 3/12/2014 | 13:20:25 | 15:29:44 | 600 | 22 | 80 | 16 | NE |
| 3/13/2014 | 10:14:14 | 13:21:29 | 500, 1100 | 25 | 80 | 7 | ENE |
| 3/14/2014 | 10:17:07 | 12:48:25 | 500, 1200, 2200, 3200 | 25 | 80 | 8 | E |
| 3/16/2014 | 10:38:18 | 13:30:33 | 500, 1000, 1200, 1600 | 25 | 80 | 5 | E |
| 3/17/2014 | 12:23:11 | 15:26:38 | 600, 1200, 1500 | 26 | 75 | 10 | ENE |
| 3/19/2014 | 10:25:33 | 13:21:48 | 600, 1600, 3200, 4800, 6500, 7100 | 22 | 95 | 7 | ENE |
| 3/21/2014 | 12:32:34 | 15:00:22 | 600, 800, 1300 | 23 | 90 | 6 | ENE |
| 3/23/2014 | 10:56:47 | 13:46:34 | 600 | 23 | 90 | 5 | NNE |
| **Dry Season** | | | | | | | |
| 9/6/2014 | 11:16:07 | 13:39:03 | 500, 1700 | 29 | N/A | 4 | ESE |
| 9/9/2014 | 11:01:14 | 14:11:01 | 500, 1600, 1900, 5800 | 25 | 80 | 4 | NE |
| 9/11/2014 | 10:32:40 | 13:37:38 | 500 | 27 | 70 | 7 | ENE |
| 9/12/2014 | 10:41:26 | 13:08:04 | 600, 1600 | 25 | 65 | 6 | E |
| 9/13/2014 | 10:50:05 | 13:43:21 | 500, 800, 1600, 2100, 2600 | 27 | 70 | 5 | ENE |
| 9/15/2014 | 10:59:06 | 14:01:47 | 500, 1800, 1900, 2600 | 28 | 60 | 5 | E |
| 9/16/2014 | 11:40:15 | 14:27:54 | 500, 1900 | 28 | 65 | 3 | NE |
| 9/18/2014 | 10:35:59 | 13:26:01 | 500, 1900, 4900 | 27 | 70 | 2 | E |
| 9/19/2014 | 10:30:23 | 13:46:44 | 500, 1600 | 28 | 70 | 2 | NE |
| 9/21/2014 | 11:17:32 | 14:20:51 | 500, 1300, 1600, 1900, 2500, 4800 | 26 | 70 | 5 | ENE |
| 9/22/2014 | 10:23:40 | 13:37:40 | 500, 1800 | 28 | 60 | 3 | NE |
| 9/23/2014 | 11:46:45 | 14:43:13 | 500, 1900 | 28 | 60 | 2 | E |
| 9/25/2014 | 13:09:24 | 15:53:59 | 500, 1600, 1800 | 28 | 70 | 4 | E |
| 9/27/2014 | 14:29:21 | 16:21:40 | 600, 2300 | 29 | 55 | 5 | ENE |
| 9/28/2014 | 11:09:12 | 14:07:27 | 600, 1800, 2100 | 28 | 65 | 5 | ENE |
| 9/30/2014 | 10:55:10 | 13:10:19 | 500, 2000, 3200 | 29 | 55 | 3 | E |
| 10/1/2014 | 10:39:01 | 13:09:54 | 600, 1000, 1300, 1600, 2100, 2500 | 26 | 65 | 6 | E |

| | | | | | | | | |
|---|---|---|---|---|---|---|---|---|
| 10/3/2014 | 10:50:51 | 13:54:57 | 800, 1000, 1200, 1600, 1900, 2600, 3300, 3900 | 26 | 75 | 4 | ESE |
| 10/4/2014 | 12:24:46 | 13:52:31 | 600 | 27 | 55 | N/A | N/A |

[a]The flight on March 10, 2014 is omitted from this table due to an AMS failure.

[b]local time

[c]altitude above mean sea level from GPS data, only level flight legs on which AMS data were collected listed

[d]immediately after takeoff, altitude < 1000 m, wind speed and heading from Aventech Research AIMMS-20 probe

T from Rosemount 102E probe, RH calculated from T and dewpoint measured by General Eastern 1011-B probe

**Table 1. Flight times, altitude of level legs, and meteorological parameters measured by instrumentation onboard the G-1 research aircraft for research flights described in this manuscript. Temperature, RH, wind speed, and wind heading correspond to values measured shortly after takeoff at altitudes less than 1000 m. Note that meteorological parameters are not homogeneous in space or time.**

| | Wet Season | Dry Season |
|---|---|---|
| **Median Loading (All Data)** | | |
| Org | 0.85 | 4.29 |
| $SO_4$ | 0.19 | 0.77 |
| $NO_3$ | 0.02 | 0.05 |
| $NH_4$ | 0.06 | 0.25 |
| **Median Loading (Altitude < 700 m)** | | |
| Org | 0.89 | 4.22 |
| $SO_4$ | 0.15 | 0.7 |
| $NO_3$ | 0.02 | 0.03 |
| $NH_4$ | 0.04 | 0.21 |
| | | |
| **Mean Loading (All data)** | | |
| Org | 0.91 | 4.41 |
| $SO_4$ | 0.16 | 0.83 |
| $NO_3$ | 0.02 | 0.1 |
| $NH_4$ | 0.05 | 0.26 |
| **Mean Loading (Altitude < 700 m)** | | |
| Org | 1.02 | 4.45 |
| $SO_4$ | 0.18 | 0.85 |

| | | |
|---|---|---|
| NO$_3$ | 0.03 | 0.05 |
| NH$_4$ | 0.06 | 0.23 |

**Table 2. Chemically resolved mass loading measured by the AMS onboard the G-1. Mass loadings are normalized to 23 °C and 1013 hPa. Units are μg/m$^3$. The statistics for altitudes below 700 m approximate the boundary layer conditions at the time of the flights.**

|  | Wet Season | Dry Season |
|---|---|---|
| **Median Concentration (all data)** | | |
| m/z 42 (acetonitrile) | 0.1 | 0.25 |
| m/z 59 (acetone) | 0.47 | 1.46 |
| m/z 69 (isoprene) | 0.43 | 1.09 |
| m/z 71 (isoprene oxidation products) | 0.66 | 1.84 |
| m/z 79 (benzene) | 0.03 | 0.08 |
| m/z 137 (monoterpenes) | 0.1 | 0.08 |
| | | |
| **Median Concentration (<700 m)** | | |
| m/z 42 (acetonitrile) | 0.09 | 0.26 |
| m/z 59 (acetone) | 0.43 | 1.65 |
| m/z 69 (isoprene) | 0.52 | 1.86 |
| m/z 71 (isoprene oxidation products) | 0.76 | 2.16 |
| m/z 79 (benzene) | 0.02 | 0.09 |
| m/z 137 (monoterpenes) | 0.12 | 0.11 |
| | | |
| **Mean Concentration (all data)** | | |
| m/z 42 (acetonitrile) | 0.12 | 0.36 |
| m/z 59 (acetone) | 0.59 | 1.91 |
| m/z 69 (isoprene) | 0.67 | 1.82 |
| m/z 71 (isoprene oxidation products) | 0.87 | 1.99 |
| m/z 79 (benzene) | 0.04 | 0.1 |
| m/z 137 (monoterpenes) | 0.12 | 0.11 |
| | | |
| **Mean Concentration (<700 m)** | | |
| m/z 42 (acetonitrile) | 0.11 | 0.4 |
| m/z 59 (acetone) | 0.7 | 2.31 |
| m/z 69 (isoprene) | 0.82 | 2.37 |
| m/z 71 (isoprene oxidation products) | 1.07 | 2.4 |
| m/z 79 (benzene) | 0.04 | 0.11 |
| m/z 137 (monoterpenes) | 0.15 | 0.14 |

**Table 3. Measurements of VOC species made by the PTR-MS onboard the G-1. Units are ppbv. The statistics for altitudes less than 700 m capture the lowest altitude data during the campaign (Table 1).**

| Number of Factors | $Q/Q_{exp}$ | Notes |
|---|---|---|
| 2 | 0.28 | 4% of mass unresolved, OOA-like factors resolved but look mixed |
| 3 | 0.26 | 2% of mass still unresolved, OOA factors looks split and unrealistic, no HOA or IEPOX SOA factor |
| 4 | 0.25 | Most of mass resolved (0.1 % residual) , HOA factor now resolved, OOA factor split, unrealistic |
| 5 | 0.24 | IEPOX factor resolved, OOA factor split, unrealistic |

**Table A1. Summary of scaled residuals and reasoning used in the choice of a 5 factor solution in the PMF analysis of the wet season HR-AMS data.**

