# Peer review of "Aircraft Observations of the Chemical Composition and Aging of Aerosol in the Manaus Urban Plume during GoAmazon 2014/5"

_Atmospheric Chemistry and Physics, 2018_

## Referee Comment (RC1) · Anonymous Referee #1 · 3 Apr 2018

Referee comments on "Aircraft Observations of Aerosol in the Manaus Urban Plume and Surrounding Tropical Forest during GoAmazon 2014/15" by Shilling et al., 2018. MS No.: acp-2018-193.

The authors have done large number of research flights over the Amazon basin. The measurements are done with state-of-the-art instrumentation (HR-AMS, PTR-MS etc). However, currently the data analysis and representation is unfortunately lacking. The authors are giving a very narrow overview of the general situation and are focusing to only one flight in more detail. I believe this dataset is important and worth publishing but that first the data presentation needs to be significantly improved.

[Figure]

Major comments: 1) The title of article is very broad. I recommend clarifying the article title to be more inline with content. Also, check that the aim of this article (defined at the end of introduction) is inline with the title and results. 2) Improve the chapter 3.1, especially figure 2 and table 1. Clarify the figure 2 to be clear and easy to understand. Maybe remove unnecessary panels. Table 1: Please add general meteorological information (like conditions, T, RH, wind during flights), also probably would be more useful to show averages over the flight legs (1-5) or average values for different heights (500m, 1000m) separately, instead of whole flight average PM composition. Also, please include the PTR-MS results to this overview (fig2, table1). 3) Why unit mass data is used in fig 2? I strongly suggest replacing all the UMR data with HR data as authors have it and it is considered to be more accurate. Assumable all other data in figures and text is HR data? Also, PMF is assumable run on HR-data? I strongly suggest using only HR data as it is available and tools are developed for the data analysis. 4) Why PMF was run only to wet season data? Why not dry season? The authors speculate that during dry season biomass burning was a major source. PMF should certainly show this. PMF for dry season would give important information on differences in PM sources between seasons also. I recommend running PMF on data from both seasons. Improved chapter 3.1 and PMF run for both seasons would help to provide a better comparison between seasons. I recommend running PMF for both seasons to do proper source analysis. 5) The aim of flights was to study evolution of Manaus plume during aging. Is it not possible to find another flight for comparison for this "golden" one? It would be interesting to see if Δorg/ΔCO would be as low and constant also in other flights. Also, would it be possible to identify any "golden" day from the dry season? The comparison in SOA aging between dry and wet season would be very interesting. Currently, as you show that the Δorg/ΔCO is quite low and does not change, it is not sure if it was only one time occurrence or the normal situation. Also, please provide some statistics how often the manaus plume was observed during the flights for wet and dry season. 6) I assume that there are a large number of publications from this campaign already published. Please provide a short summary about those and how this differs from those
previously published. 7) Please check that all abbreviations (PTR-MS, HR-ToF-AMS, G-1 etc) are explained when first mentioned.

Detailed comments:

Page 1, Line 31. Please define "golden" Page 2, Line 24. Define Lagrangian Page 4, line 25. The aim of this study could be better defined. Especially the scientific aim. Page 5, line 15. Explain why 13 seconds time interval was chosen? AMS can be used with much better time resolution (Hz), especially when ptof is not used. Page 5, line 25. Define the software models used for the data analysis (igor version+ Squirrel and PIKA versions+ PMF software). Also, please add details about PMF analysis (number of factors tested, how nr of factors was decided, were factors constrained, ME2 etc.). The main parameters should be in the manuscript also, since not many people does not read the supplement. Page 5, line 25-26. Please explain why data was normalized to 23C and 1013hPA ? Page 5, line 28. Please provide the model and manufacturer for the PTR-MS and all other instruments. Page 6. Chapter 3.1. Please add a description of local conditions (temp, RH, wind etc) during the wet and dry periods. Also, provide info about the flight altitudes, times etc. It is hard to compare the results when it is not known if the difference is due to altitude or time of day or due to different source. Are the points in fig 2. top panel average values over the flights (including all altitudes?)? were all flights similar? Page 6, line 20-30/table 1. Are all these concentrations above the AMS detection limit? Especially the ammonium tends to have higher detection limits in aerosol mass spectrometers.. Page 7. Line 15-16. Is there any proof for this? High CO2, CO, BC or levoglucosan values? Or tracers of levoglucosan (HR ions at m/z 63,70) at mass spectra? Page 8. The title of chapter 3.2 (Case Study, March 13, 2014 Flight) is not very descriptive. Consider changing it to something that describes content. Page 11. Line 30. How the age of plume was estimated to be 4-5 hours? Page 11. Please define "photochemical clock" Page 13. Chapter 3.3. I am confused. This is not based on the case study, but on all flights where manaus plume was observed? Page 13, row 11, 25. Please replace " SO4 " to sulphate.

[Figure]

Figures

Fig 1. Are these T1-T3 ground stations used in this article? Fig 2. All the panels are showing same thing. Maybe consider how this figure could be condensed to a smaller figure? Also, the connecting lines between the average values are misleading. Maybe mark the median values with different colors. Also in top plot, some of the points (e.g. point between feb 27 and Mar 04) seems to maybe overlap? Fix these please. Fig 5. Legends from bottom panel are missing

[Figure]

---

## Referee Comment (RC2) · Anonymous Referee #2 · 8 Apr 2018

This manuscript reports observations from an aircraft study over Manaus, Brazil. The discussions mainly focus on the chemical evolution of aerosol particles in the Manaus plume sampled on March 13, 2014 as it was transported to the surrounding and nearby Amazon tropical rainforest. A particularly interesting observation is that $\Delta$org/$\Delta$CO ratio stayed nearly constant in the Manaus urban plume although OA oxidation increased continuously during aging. The G1 measurement data from the GoAmazon campaign are very rich and this manuscript provides new and timely information on aerosol particles from an important, but poorly studied, environment. The finding of no net SOA formation in fresh urban emissions that is different from most published results on urban outflows is unique and may motivate future studies. Overall, this manuscript is

worthy of publication and ACP is a suitable journal for it. Following are some detailed comments.

The overview of the G-1 aerosol data is a bit cursory. The discussions could be expanded a bit more and other measurement results are incorporated to give a more in depth view about the atmospheric characteristics and seasonal differences. For example, the authors could consider adding other measurement data on Fig 2 and discuss them in connection with the AMS aerosol data. In addition, in section 3.1, more background information on the dry and wet seasons may be necessary to establish the purpose and the significance of the comparison.

More detailed diagnostic information should be presented to confirm the PMF results.

For the finding of constant $\Delta org/\Delta CO$ and increasing aerosol oxidation in an aging urban plumes, one question is the behaviors of VOCs according to PTR-MS measurements? Was the March 13 flight an isolated case or is the phenomenon more general for the Manaus emissions? What are the $\Delta org$ vs $\Delta CO$ values for other flights? How do they compare? Are there seasonal differences?

1. Page 5, line 15, what does the "13s data sampling interval" correspond to? was the AMS operated under the standard MS mode (equal chopper-open and closed positions) or the fast sampling mode typically used for aircraft sampling?

2. Page 5, line 28, what's the m/z range for the PTR-MS measurement?

3. Page 6, line 1, define E/N.

4. Page 6, what's SAMBBA?

5. Page 6, line 31, mention the year and month for the Chen et al. study.

6. Page 7, Line 13, are there diurnal mixed layer height data to justify the usage of 700 m?

7. Page 7, line 18, what does 'sources' mean in this context?

[Figure]

8. Page 7, line 22, what does "aircraft product distribution' mean?

9. Page 7 line 24, "suggests"

10. Page 7, line 30 - 31, was biomass burning influence detected during this study? What are the evidences? The citations were clearly not from Go-Amazon 2014/15.

11. Page 7, line 32, spell out MSA.

12. Page 8, line 1-2, give citation.

13. How well is the correlation between IEPOX-SOA and m/z 82?

14. Page 10, line 5, "this date"

15. Page 10, line 15 – 16, might be useful to report the correlation coefficients.

16. Page 10, line 18, clarify the meaning of "transformation of HOA to SOA/OOA", e.g., through what mechanisms. Is oxidized POA counted as SOA?

17. Page 10, line 23 - 24, this sentence is vague. What chemical mechanism?

18. Page 10, line 26, limit of detection for IEPOX SOA was mentioned. What's the value and how was it determined?

19. Page 11, line 5 -9, how was $\Delta org/\Delta CO$ determined, through background subtraction or linear fit? Either way, please provide details, e.g., the choice of background values or quality of linear fit etc. Consider to move some information in the supplementary to main body of the manuscript.

20. Page 12, line 14 – 15, change "occurring" to "is occurring".

21. Page 13, line 18, speaking of sulfur emissions and oxidation of compounds such as DMS, it would be necessary to check for the presence of methylate /MSA in particles. BTW, DMS should be defined.

22. Page 14, line 5, it is not that aerosol composition "did not change" but rather "did

not change significantly".

23. Figure 1, consider to mention in the figure caption the duration of the flight.

24. Figure 2, how were the average pie charts calculated, straight averages of the fractions or mass weighted averages?

25. Figure 4, What are the aerosol species concentrations, same as mentioned in Fig. 2 (23C and 1 atm)?

26. According to Figure 4, the time series of O3 and CO appear to correlate quite well. Figure 5 shows that HOA and OOA go up and down together as well. It would be interesting to know the cross correlations between OOA, HOA, CO and O3.

27. Figure 5, was NO2 measured? How does Ox time series look like?

28. Figure 6, consider to add error bars for the data.

———————————————————

---

## Author Comment (AC1) · 26 Jun 2018

We thank the reviewer for the comments and feel they will improve the manuscript. In response to these comments, we have made substantial changes to the manuscript. Below, we respond to the reviewers comments in detail and indicate changes we have made to the manuscript. Referee comments are in black, plain text. Our response to referees is in black, *italic* text. Changes to the text are listed in blue text.

Referee #1

Referee comments on "Aircraft Observations of Aerosol in the Manaus Urban Plume

and Surrounding Tropical Forest during GoAmazon 2014/15" by Shilling et al., 2018.

MS No.: acp-2018-193.

The authors have done large number of research flights over the Amazon basin. The measurements are done with state-of-the-art instrumentation (HR-AMS, PTR-MS etc). However, currently the data analysis and representation is unfortunately lacking. The authors are giving a very narrow overview of the general situation and are focusing to only one flight in more detail. I believe this dataset is important and worth publishing but that first the data presentation needs to be significantly improved.

Major comments: 1) The title of article is very broad. I recommend clarifying the article title to be more inline with content. Also, check that the aim of this article (defined at the end of introduction) is inline with the title and results.

*We have updated the title. We have also revised the last portion of the introduction.*

Aircraft Observations of the Chemical Composition and Aging of Aerosol in the Manaus Urban Plume during GoAmazon 2014/5

In this manuscript we report on measurements from instruments deployed on the G-1 focusing primarily on measurements of aerosol species and trace gases that impact the aerosol lifecycle. In the first part of the manuscript, we provide an overview of aerosol and VOC measurements and compare and contrast the wet and dry season data. In the second portion of the manuscript, we examine, in detail, the first 4 – 6.5 hours of photochemical aging of the Manaus plume as it is transported into the surrounding tropical forest and interacts with biogenic emissions.

2) Improve the chapter 3.1, especially figure 2 and table 1. Clarify the figure 2 to be clear and easy to understand. Maybe remove unnecessary panels. Table 1: Please add general meteorological information (like conditions, T, RH, wind during flights), also probably would be more useful to show averages over the flight legs (1-5) or average values for different heights (500m,1000m) separately, instead of whole flight average PM composition. Also, please include the PTR-MS results to this overview (fig2, table1).

*In regard to Figure 1, please see our response to a similar comment below.*

*We have added an additional table (Table 1) summarizing the flight conditions the reviewer suggest. We have added text pointing to Table 1 in the text.*

Table 1 lists takeoff and landing times, the altitude of level flight legs, and meteorological parameters measured by the G-1 instrumentation shortly after takeoff on each flight. Flights generally departed in the late morning or early afternoon and lasted between 3 and 3.5 hours (Table 1).

| Flight Date[a] | Takeoff [b] | Landing [b] | Altitude(s) of Level Flight Legs (m)[c] | T (°C)[d] | RH (% water)[d] | Wind Speed (m/s)[d] | Wind Heading[d] |
|---|---|---|---|---|---|---|---|
| **Wet Season** | | | | | | | |
| 2/22/2014 | 10:37:11 | 13:25:28 | 600, 1600 | 25 | 80 | 5 | ENE |
| 2/25/2014 | 12:30:31 | 14:43:04 | 600, 1300 | 26 | 80 | 8 | ENE |
| 3/1/2014 a | 9:33:09 | 11:31:31 | 600 | 25 | 80 | 6 | NE |
| 3/1/2014 b | 13:13:59 | 14:49:34 | 1100, 4500, 6400 | 24 | 80 | 4 | NE |
| 3/3/2014 | 13:46:34 | 15:11:57 | 600 | 26 | 90 | N/A | N/A |
| 3/7/2014 | 9:08:11 | 11:39:18 | 600, 1300, 1900, 3200, 3800, 4500, 5800 | 26 | 65 | 10 | NNE |
| 3/11/2014 | 10:39:59 | 13:51:10 | 600, 1500 | 25 | 85 | 7 | ENE |
| 3/12/2014 | 13:20:25 | 15:29:44 | 600 | 22 | 80 | 16 | NE |
| 3/13/2014 | 10:14:14 | 13:21:29 | 500, 1100 | 25 | 80 | 7 | ENE |
| 3/14/2014 | 10:17:07 | 12:48:25 | 500, 1200, 2200, 3200 | 25 | 80 | 8 | E |
| 3/16/2014 | 10:38:18 | 13:30:33 | 500, 1000, 1200, 1600 | 25 | 80 | 5 | E |
| 3/17/2014 | 12:23:11 | 15:26:38 | 600, 1200, 1500 | 26 | 75 | 10 | ENE |
| 3/19/2014 | 10:25:33 | 13:21:48 | 600, 1600, 3200, 4800, 6500, 7100 | 22 | 95 | 7 | ENE |
| 3/21/2014 | 12:32:34 | 15:00:22 | 600, 800, 1300 | 23 | 90 | 6 | ENE |
| 3/23/2014 | 10:56:47 | 13:46:34 | 600 | 23 | 90 | 5 | NNE |
| **Dry Season** | | | | | | | |
| 9/6/2014 | 11:16:07 | 13:39:03 | 500, 1700 | 29 | N/A | 4 | ESE |
| 9/9/2014 | 11:01:14 | 14:11:01 | 500, 1600, 1900, 5800 | 25 | 80 | 4 | NE |
| 9/11/2014 | 10:32:40 | 13:37:38 | 500 | 27 | 70 | 7 | ENE |
| 9/12/2014 | 10:41:26 | 13:08:04 | 600, 1600 | 25 | 65 | 6 | E |
| 9/13/2014 | 10:50:05 | 13:43:21 | 500, 800, 1600, 2100, 2600 | 27 | 70 | 5 | ENE |
| 9/15/2014 | 10:59:06 | 14:01:47 | 500, 1800, 1900, 2600 | 28 | 60 | 5 | E |
| 9/16/2014 | 11:40:15 | 14:27:54 | 500, 1900 | 28 | 65 | 3 | NE |
| 9/18/2014 | 10:35:59 | 13:26:01 | 500, 1900, 4900 | 27 | 70 | 2 | E |
| 9/19/2014 | 10:30:23 | 13:46:44 | 500, 1600 | 28 | 70 | 2 | NE |
| 9/21/2014 | 11:17:32 | 14:20:51 | 500, 1300, 1600, 1900, 2500, 4800 | 26 | 70 | 5 | ENE |

| | | | | | | | |
|---|---|---|---|---|---|---|---|
| 9/22/2014 | 10:23:40 | 13:37:40 | 500, 1800 | 28 | 60 | 3 | NE |
| 9/23/2014 | 11:46:45 | 14:43:13 | 500, 1900 | 28 | 60 | 2 | E |
| 9/25/2014 | 13:09:24 | 15:53:59 | 500, 1600, 1800 | 28 | 70 | 4 | E |
| 9/27/2014 | 14:29:21 | 16:21:40 | 600, 2300 | 29 | 55 | 5 | ENE |
| 9/28/2014 | 11:09:12 | 14:07:27 | 600, 1800, 2100 | 28 | 65 | 5 | ENE |
| 9/30/2014 | 10:55:10 | 13:10:19 | 500, 2000, 3200 | 29 | 55 | 3 | E |
| 10/1/2014 | 10:39:01 | 13:09:54 | 600, 1000, 1300, 1600, 2100, 2500 | 26 | 65 | 6 | E |
| 10/3/2014 | 10:50:51 | 13:54:57 | 800, 1000, 1200, 1600, 1900, 2600, 3300, 3900 | 26 | 75 | 4 | ESE |
| 10/4/2014 | 12:24:46 | 13:52:31 | 600 | 27 | 55 | N/A | N/A |

[a]The flight on March 10, 2014 is omitted from this table due to an AMS failure.

[b]local time

[c]altitude above mean sea level from GPS data, only level flight legs on which AMS data were collected listed

[d]immediately after takeoff, altitude < 1000 m, wind speed and heading from Aventech Research AIMMS-20 probe

T from Rosemount 102E probe, RH calculated from T and dewpoint measured by General Eastern 1011-B probe

**Table 1. Flight times, altitude of level legs, and meteorological parameters measured by instrumentation onboard the G-1 research aircraft for research flights described in this manuscript. Temperature, RH, wind speed, and wind heading correspond to values measured shortly after takeoff at altitudes less than 1000 m. Note that meteorological parameters are not homogeneous in space or time.**

*Table 2 (formerly Table 1) compares the average and median aerosol mass loading of the low altitude legs to the same quantities averaged over all altitudes. We wish to focus the discussion on the evolution of the Manaus plume and therefore chose to highlight the lowest altitude data in Table 1. All flights did not follow identical flight patterns, so segregating into flight legs is not possible. A visualization of all the flight patterns is shown in Martin et al., 2016.*

*We have added a figure and table showing the PTR-MS data at the reviewer's suggestion. We have incorporated a discussion of this data into the text.*

[Figure]

|  | Wet Season | Dry Season |
|---|---|---|
| **Median Concentration (all data)** | | |
| m/z 42 (acetonitrile) | 0.1 | 0.25 |
| m/z 59 (acetone) | 0.47 | 1.46 |
| m/z 69 (isoprene) | 0.43 | 1.09 |
| m/z 71 (isoprene oxidation products) | 0.66 | 1.84 |
| m/z 79 (benzene) | 0.03 | 0.08 |
| m/z 137 (monoterpenes) | 0.1 | 0.08 |
| | | |
| **Median Concentration (<700 m)** | | |
| m/z 42 (acetonitrile) | 0.09 | 0.26 |
| m/z 59 (acetone) | 0.43 | 1.65 |
| m/z 69 (isoprene) | 0.52 | 1.86 |
| m/z 71 (isoprene oxidation products) | 0.76 | 2.16 |

| | | |
|---|---|---|
| m/z 79 (benzene) | 0.02 | 0.09 |
| m/z 137 (monoterpenes) | 0.12 | 0.11 |

**Mean Concentration (all data)**

| | | |
|---|---|---|
| m/z 42 (acetonitrile) | 0.12 | 0.36 |
| m/z 59 (acetone) | 0.59 | 1.91 |
| m/z 69 (isoprene) | 0.67 | 1.82 |
| m/z 71 (isoprene oxidation products) | 0.87 | 1.99 |
| m/z 79 (benzene) | 0.04 | 0.1 |
| m/z 137 (monoterpenes) | 0.12 | 0.11 |

**Mean Concentration (<700 m)**

| | | |
|---|---|---|
| m/z 42 (acetonitrile) | 0.11 | 0.4 |
| m/z 59 (acetone) | 0.7 | 2.31 |
| m/z 69 (isoprene) | 0.82 | 2.37 |
| m/z 71 (isoprene oxidation products) | 1.07 | 2.4 |
| m/z 79 (benzene) | 0.04 | 0.11 |
| m/z 137 (monoterpenes) | 0.15 | 0.14 |

**Table 3. Measurements of VOC species made by the PTR-MS onboard the G-1. Units are ppbv. The statistics for altitudes less than 700 m capture the lowest altitude data during the campaign (Table 1).**

While this manuscript will focus on the particulate data, measurements of the volatile organic compounds provide insights into the precursors that are oxidized to form OA and help to identify the source of an air parcel. Figure 3 and Table 3 summarize the concentrations of several relevant VOCs measured on board the G-1 with the PTR-MS. Similar to the trends seen in the aerosol mass loading data, concentrations of most VOCs measured by the PTR-MS are significantly higher in the dry season than in the wet season. Concentrations of isoprene and its oxidation products, biogenic precursors of OA, are a factor of 2-3 times higher in the dry season than the wet season. The average daily high temperatures in Manaus are 33.5°C in September (dry season) and 30.9°C in March (wet season) and isoprene emissions have been shown to scale with temperature, among other variables (Guenther et al., 2012). Measurements of benzene, which is primarily anthropogenic in origin, were also significantly higher in the dry season. While benzene itself is unlikely to significantly contribute to SOA formation over the timescales we observe on the G-1 flights (4-6.5 hours) due to its ~5 day atmospheric lifetime (Atkinson and Arey, 2003), other unmeasured anthropogenic VOC concentrations which would contribute to more immediate OA formation may have been higher as well. The higher VOC concentrations may contribute to the higher OA concentrations measured in the dry season. However,

the PTR-MS data, in addition to satellite and remote sensing measurements (Martin et al., 2016), also show that biomass burning significantly impacted the region. Measurements of acetonitrile, whose major source is biomass burning (Yokelson et al., 2009;Yokelson et al., 2007), are 2-3 times higher in the dry season than the wet season, indicating a significant biomass burning impact in the region. Biomass burning is a known source of OA (Jolleys et al., 2012;Bond et al., 2004;Yokelson et al., 2009;Ferek et al., 1998) and would also contribute to the higher aerosol concentrations observed in the dry season.

3) Why unit mass data is used in fig 2? I strongly suggest replacing all the UMR data with HR data as authors have it and it is considered to be more accurate. Assumable all other data in figures and text is HR data? Also, PMF is assumable run on HR-data? I strongly suggest using only HR data as it is available and tools are developed for the data analysis.

*We have replaced the use of UMR data with HR data in Figure 2 and Table 1. In the original manuscript PMF was run using HR data. All data presented in the paper now use HR data analysis.*

All AMS data in the manuscript have been processed using the high-resolution data analysis routine.

4) Why PMF was run only to wet season data? Why not dry season? The authors speculate that during dry season biomass burning was a major source. PMF should certainly show this. PMF for dry season would give important information on differences in PM sources between seasons also. I recommend running PMF on data from both seasons. Improved chapter 3.1 and PMF run for both seasons would help to provide a better comparison between seasons. I recommend running PMF for both seasons to do proper source analysis.

*PMF analysis is in progress for the dry season. A full description of the PMF analysis results will be the subject of a future publication. For this reason, we have also refrained from describing the PMF results for the entire wet season dataset and only present the PMF analysis on the March 13 flight. Our goal in section 3.1 is not to do a full and comprehensive source analysis, but to present a high-level overview of the observations in the wet and dry season. We have added references for the impact of biomass burning on the aerosol concentrations throughout the manuscript.*

5) The aim of flights was to study evolution of Manaus plume during aging. Is it not possible to find another flight for comparison for this "golden" one? It would be interesting to see if Δorg/ΔCO would be as low and constant also in other flights.

*At the reviewer's suggestion, we have screened the data and analyzed additional flights during the wet season. We selected flights on which there was: a clear contrast between the plume and background data, data collected in the boundary layer, a complete dataset from key instrument (AMS, CO detector), minimal interference from biomass burning emissions and cloud processing, and a flight plan extending a similar downwind distance. We were able to identify two additional flights that can be presented, March 11 and March 16. Additional flights Both show similar trends in the data with Δorg/ΔCO values averaging $34 \mu g/m^3 \ ppmv^{-1}$ and showing little trend with aging. We have added a figure to the manuscript showing this additional data and have added a discussion of this data to the manuscript. In addition, as we cite in the manuscript, and analysis of data at the T2 and T3 ground sites by Cirino et al., 2018 shows no*

*change in Δorg/ΔCO from T2 to T3. The data thus suggest the behavior is representative for the wet season. Of course, repeated measurements over several years would be valuable to assess whether 2014 was a typical or atypical wet season.*

Thus far, we have focused on a detailed analysis of the March 13, 2014 plume because there is a clear contrast between the background and plume data through the portion of the flight in the boundary layer, there is little interference from biomass burning and cloud processing, the flight pattern was extended far enough downwind to investigate a significant aging time, and the data from all key instruments are available and complete. In an effort to understand whether the Δorg/ΔCO observations on March 13 are representative of plume aging in the wet season, we have screened the rest of the wet season flight data and, based on the criteria mentioned above, we have identified two additional flights, March 11 and March 16, 2014, for analysis. The Δorg/ΔCO values for these data as a function of distance from Manaus are shown in Figure 8. On March 11, Δorg/ΔCO values average 29.9 $\mu g/m^3$ $ppmv^{-1}$, show little change with distance from Manaus, and show similar variability as the March 13 flight data. The mean wind speed for the analyzed portion of the flight was 5.3 m/s translating to an approximate maximum plume age of 5-5.5 hours. On March 16, Δorg/ΔCO values average 42.0 $\mu g/m^3$ $ppmv^{-1}$, show little significant increase with distance from Manaus, and exhibit a larger range of values than the other two datasets. The mean wind speed for the analyzed portion of the March 16 flight was 4.6 m/s translating to an approximate maximum plume age of 6-6.5 hours. We estimate that the error in the Δorg/ΔCO measurement is approximately 10 $\mu g/m^3$ $ppmv^{-1}$, thus the difference in average Δorg/ΔCO values between the March 16 and the March 11 and 13 datasets are at the edge of what we would consider significant. The average Δorg/ΔCO is 34.3 $\mu g/m^3$ $ppmv^{-1}$ for all flights and collectively represent the first 4 – 6.5 hours of aging. In addition to the aircraft data, Cirino et al. (2018) report that the median Δorg/ΔCO values for the plume as measured at the T2 and T3 sites are nearly identical to one another in the wet season and similar to measurements from the G-1(Cirino et al.). Thus, the data from both G-1 and the ground sites suggests that Δorg/ΔCO values of 30-40 $\mu g/m^3$ $ppmv^{-1}$ that change little in the first 4-6.5 hours of plume aging are representative of the Manaus plume behaviour in the wet season.

[Figure]

Also, would it be possible to identify any "golden" day from the dry season? The comparison in SOA aging between dry and wet season would be very interesting. Currently, as you show that the $\Delta org/\Delta CO$ is quite low and does not change, it is not sure if it was only one time occurrence or the normal situation.

*The dry season data are complicated by the presence of a regional (and sometimes local) biomass burning background. Understanding and separating the evolution of the Manaus urban plume from evolution of the biomass burning is a complex task and will be the focus of a future publication. In the current manuscript, we want to focus on the evolution of the plume in the wet season since this represents, clean background conditions. We show in the manuscript that these observation are unique relative to that is typically observed in other parts of the world. Adding detailed information on the evolution of biomass burning plumes is a separate topic and would warrant a separate publication.*

Also, please provide some statistics how often the manaus plume was observed during the flights for wet and dry season.

*In the wet season, we encounter the Manaus plume between 15 – 25% of the total flight time. In the dry season, we encounter the Manaus plume between 15 – 35% of the total flight time. These statistics are much more uncertain for the dry season. In the dry season, we frequently encounter biomass burning plumes, and it is at times difficult to differentiate a biomass burning plume from the Manaus urban plume, particularly when they mix.*

6) I assume that there are a large number of publications from this campaign already published. Please provide a short summary about those and how this differs from those previously published.

*The goals of GoAmazon were very broad. Summarizing all publications form GoAmazon would be beyond the scope of this work and would introduce information that is not directly relevant to this manuscript. We believe we have adequately referenced both past work in the Amazon region and the GoAmazon 2014/5 publications that are relevant to this study, particularly those investigating the impact of the Manaus plume on SOA, in the text. In total, we cite and discuss 14 GoAmazon manuscripts. We feel it is better to discuss these manuscripts by subject as we have done, rather than as a separate section on GoAmazon 2014/5 as the reviewer suggests. Currently, there are no other publications in the literature describing the G-1 AMS measurements from GoAmazon 2014/5 or the evolution and aging of the Manaus plume, so this manuscript is unique.*

7) Please check that all abbreviations (PTR-MS, HR-ToF-AMS, G-1 etc.) are explained when first mentioned.

*We have checked these abbreviations and spelled them out the first time they are mentioned.*

As part of this campaign, the DOE Gulfstream-1 (G-1) research aircraft

An Aerodyne High-Resolution Time of Flight Aerosol Mass Spectrometer (abbreviated as AMS hereafter) was deployed on

An Ionicon quadrupole high-sensitivity Proton Transfer Reaction Mass Spectrometer (PTR-MS) was used to measure selected

Detailed comments:

Page 1, Line 31. Please define "golden"

*"Golden day" is colloquialism used in discussing field studies to indicate an exemplar day. We have replace golden with exemplar.*

In the second portion of the manuscript, we discuss the evolution of the Manaus plume on March 13, 2014, one of the exemplar days in the wet season.

Page 2, Line 24. Define Lagrangian.

*Lagrangian evolution here refers to measuring the properties of the same air parcel as it evolves. We have added a brief discussion of Lagrangian sampling in the text. Additional information can be found in: (Seinfeld, 1998;Jacob, 1999).*

For several years, there has been an interest in studying the Lagrangian (i.e., within the same air parcel) evolution of organic aerosol from the emissions of urban centers.

Most studies of this type are best described as pseudo-Lagrangian as repeatedly sampling the same air parcel is difficult with mobile platforms and impossible with fixed sites. Dilution and mixing of the air parcel with background air also alter the plume composition.

Page 4, line 25. The aim of this study could be better defined. Especially the scientific aim.

*We have revised lines 24-25.*

In this manuscript we report on measurements from instruments deployed on the G-1 focusing primarily on measurements of aerosol species and trace gases that impact the aerosol lifecycle. In the first part of the manuscript, we provide an overview of aerosol and VOC measurements and compare and contrast the wet and dry season data. In the second portion of the manuscript, we examine, in detail, the first 4 – 6.5 hours of photochemical aging of the Manaus plume as it is transported into the surrounding tropical forest and interacts with biogenic emissions.

Page 5, line 15. Explain why 13 seconds time interval was chosen? AMS can be used with much better time resolution (Hz), especially when ptof is not used.

*The sampling time was chosen as a compromise between adequate time resolution and achieving adequate signal, particularly for the wet season where aerosol loadings were low. The need for higher S/N outweighed the need for higher time resolution, which we deemed adequate. See also our response to a similar question from Reviewer 2.*

Page 5, line 25. Define the software models used for the data analysis (igor version+ Squirrel and PIKA versions+ PMF software). Also, please add details about PMF analysis (number of factors tested, how nr of factors was decided, were factors constrained, ME2 etc.).The main parameters should be in the manuscript also, since not many people does not read the supplement.

*We have added the software versions used in the analysis to the manuscript.*

…using the PMF Evaluation Tool (v 2.06) and the PFM2 algorithm (v 4.2)…

Data was analyzed in Igor Pro (v6.37) using the high-resolution analysis package (Squirrel v1.55H, PIKA v1.44H) and techniques described in the literature (Canagaratna et al., 2015;Kroll et al., 2011;Aiken et al., 2007;Allan et al., 2003;Jimenez et al., 2003). All AMS data in the manuscript have been processed using the high-resolution data analysis routine. The O:C and H:C values reported here use the updated calibrations described in Canagaratna et al. (2015).

*We have added details on the PMF analysis and on choosing the appropriate number of factors. We moved the discussion of PMF to the appendix at the reviewer's suggestion. The changes made to the manuscript are detailed in a response similar comment from Reviewer #2 (below).*

Page 5, line 25-26. Please explain why data was normalized to 23C and 1013 hPA?

*These are the laboratory conditions under which the flow calibration was performed. We have added this to the manuscript.*

All AMS data are normalized to the laboratory calibration conditions of 23 °C 1013 hPa.

Page 5, line 28. Please provide the model and manufacturer for the PTR-MS and all other instruments.

*The text states:* "An Ionicon quadrupole high-sensitivity PTR-MS was used to measure selected gas-phase volatile organic carbon (VOC) concentrations (Lindinger et al., 1998)" *The manufacturer is*

*Ionicon and they specify the model as a high-sensitivity PTR-MS. Further information on all instruments deployed on the G-1 is available in the supplemental information of Martin et al., 2016. We have added a reference to Martin et al. 2016.*

Additional information on the instrumentation deployed on the G-1 can be found in the supplementary information of Martin *et al.*, 2016.

Page 6. Chapter 3.1. Please add a description of local conditions (temp, RH, wind etc) during the wet and dry periods. Also, provide info about the flight altitudes, times etc. It is hard to compare the results when it is not known if the difference is due to altitude or time of day or due to different source. Are the points in fig 2. top panel average values over the flights (including all altitudes?)?were all flights similar?

*At the reviewer's suggestion, we have added a table listing meteorological conditions (Table 1) during each flight and referenced this table in the text. It is important to note that each flight samples a wide range of T, RH, wind speed, wind direction values. The listed values are for the lowest flight legs near the start of the flight. We have also added a brief overview of the meteorology in the wet and dry seasons.*

*The top panel of Figure 2 is a box and whisker plot showing the 10, 25, 50, 75, and 90[th] percentile values for each flight. Each box and whisker summarizes all the data for a particular flight, including all altitudes.*

The timing of these flight missions was chosen to provide a contrast between the wet and dry seasons (Martin et al., 2016). In the wet season, back trajectory analysis indicated that the Manaus region was typically under the influence of air originating from the North Atlantic Ocean (Martin et al., 2016). Regular, organized mesoscale convective systems triggered by sea breeze circulation brought widespread, moderate rate precipitation to the region (Giangrande et al., 2017;Machado et al., 2018;Burleyson et al., 2016). The high level of rainfall and moisture in the wet season inhibited biomass burning and a low number of fires were observed (Martin et al., 2016;Machado et al., 2018). Under these conditions, the Amazon basin is one of the cleanest continental regions on Earth onto which the Manaus plume represents a significant perturbation (Martin et al., 2010;Artaxo et al., 2013). In the dry season, back trajectory analysis indicate air masses originate from the Southern Hemisphere and travel up the Amazon River transporting pollution from the northern coastal cities (Martin et al., 2016). In addition, recirculation events transported air from the southern Amazon basin into the Manaus region (Martin et al., 2016). Intense biomass burning fires in the central and southern portion of the Amazon basin and in central Africa were observed in the dry season and a portion of these emissions were transported to the Manaus region (Martin et al., 2016;Artaxo et al., 2013;Martin et al., 2010). In the dry season, more intense but less frequent convection produced approximately one quarter of the total rainfall observed in the wet season. As a result of the combination of transport and precipitation frequency, the Manaus region is significantly more polluted in the dry season.

Page 6, line 20-30/table 1. Are all these concentrations above the AMS detection limit? Especially the ammonium tends to have higher detection limits in aerosol mass spectrometers.

*Based methodology described in DeCarlo et al., 2006, we calculate the detection limits of the AMS at the 13s averaging interval as 0.12, 0.01, 0.014, and 0.005 for organics, sulfate, nitrate,*

*and ammonium. We have added these detection limits to the manuscript. The values reported in Table 1 are above the detection limits.*

Based on the standard deviation of these blank measurements (3σ) as described in the literature (DeCarlo et al., 2006), the detection limit of the AMS at the 13s sampling interval were 0.12, 0.01, 0.014, and 0.005 μg/m³ for organics, sulfate, nitrate, and ammonium respectively.

Page 7. Line 15-16. Is there any proof for this? High CO2, CO, BC or levoglucosan values? Or tracers of levoglucosan (HR ions at m/z 63,70) at mass spectra?

*We have added references in multiple locations in the text for the presence of biomass burning in the dry season. PTR-MS measurements of acetonitrile show an influence of biomass burning, as do measurements of CO. The changes to the manuscript are provided in our response to a similar comment from Reviewer #2 (below).*

Page 8. The title of chapter 3.2 (Case Study, March 13, 2014 Flight) is not very descriptive. Consider changing it to something that describes content.

*We have changed the heading to be more descriptive.*

3.2 Evolution of Organic Aerosol in the Manaus Plume

Page 11. Line 30. How the age of plume was estimated to be 4-5 hours?

*The text states:* "Based on the mean wind speeds observed along the flight track (7.3 m s⁻¹) and the transport distance (up to 100 km), we estimate that the plume was 4-5 hours old at the farthest leg and freshly emitted over the city " . *The age of the plume was estimated from the transport distance and the measured mean wind speed along the flight track. The plume age was calculated using the equation: distance=wind speed*time.*

Page 11. Please define "photochemical clock"

*A photochemical clock is a method of estimating the photochemical age of an air parcel based on measuring the ratios of two chemicals which are photochemically removed from the atmosphere at different and known rates. More details can be found in the following references, which have been added to the manuscript (de Gouw et al., 2005;Kleinman et al., 2003;Parrish et al., 1992).*

Unfortunately, photochemical clocks could not be used to more precisely calculate the photochemical age of the plume (de Gouw et al., 2005;Kleinman et al., 2003;Parrish et al., 1992).

Page 13. Chapter 3.3. I am confused. This is not based on the case study, but on all flights where manaus plume was observed?

*We have specified in the text when we are specifically discussing the March 13 flight and when we are discussing the data in general.*

Page 13, row 11, 25. Please replace " SO4 " to sulphate.

*We have made the suggested change.*

Figures

Fig 1. Are these T1-T3 ground stations used in this article?

*Only data from the aircraft are directly analyzed in this manuscript. Data collected at the T2 and T3 sites are referenced in the article. The aircraft data are compared to the ground site data in the manuscript, particularly those collected at the T3 site.*

Fig 2. All the panels are showing same thing. Maybe consider how this figure could be condensed to a smaller figure? Also, the connecting lines between the average values are misleading. Maybe mark the median values with different colors. Also in top plot, some of the points (e.g.point between feb 27 and Mar 04) seems to maybe overlap? Fix these please.

*We have fixed the slight overlap of the points.*

*We disagree with the reviewer's statement that all the panels show the same thing. The top panel shows box and whisker plots of the absolute mass loadings for each flight. The middle panel shows the relative non-refractory PM composition averaged on a per-flight basis. The bottom panel shows the average non-refractory PM composition of all flights in each season. Thus, one panel shows absolute mass while the other two show different versions of relative mass. The most similar panels are the lower two and we feel there is value in showing the average non-refractory PM composition both on a per-flight basis and as season-averages. These types of figures are common in the literature and we prefer to keep them.*

*The figure caption states that the lines are shown to guide the eye. We disagree that the lines connecting the median values are misleading and the reviewer isn't clear on why they think so. Nevertheless, we have changed them from solid lines to dotted lines to assuage some of the reviewer's concern. We feel the lines are useful, particularly for the inorganic species which are present in much lower concentrations and thus harder to see.*

Fig 5.Legends from bottom panel are missing

*We have added a legend to the bottom panel.*

Referee #2

This manuscript reports observations from an aircraft study over Manaus, Brazil. The discussions mainly focus on the chemical evolution of aerosol particles in the Manaus plume sampled on March 13, 2014 as it was transported to the surrounding and nearby Amazon tropical rainforest. A particularly interesting observation is that $\Delta$org/$\Delta$CO ratio stayed nearly constant in the Manaus urban plume although OA oxidation increased continuously during aging. The G1 measurement data from the GoAmazon campaign are very rich and this manuscript provides new and timely information on aerosol particles from an important, but poorly studied, environment. The finding of no net SOA formation in fresh urban emissions that is different from most published results on urban outflows is unique and may motivate future studies. Overall, this

manuscript is worthy of publication and ACP is a suitable journal for it. Following are some detailed comments.

The overview of the G-1 aerosol data is a bit cursory. The discussions could be expanded a bit more and other measurement results are incorporated to give a more in depth view about the atmospheric characteristics and seasonal differences. For example, the authors could consider adding other measurement data on Fig 2 and discuss them in connection with the AMS aerosol data. In addition, in section 3.1, more background information on the dry and wet seasons may be necessary to establish the purpose and the significance of the comparison.

*We have added Figure 3 and Table 3 showing the PTR-MS data and text discussing this data in reponse to this comment and a similar comment from Reviewer #1. See also our response to a similar comment from Reviewer #1.*

While this manuscript will focus on the particulate data, measurements of the volatile organic compounds provide insights into the precursors that are oxidized to form OA and help to identify the source of an air parcel. Figure 3 and Table 3 summarize the concentrations of several relevant VOCs measured onboard the G-1 with the PTR-MS. Similar to the trends seen in the aerosol mass loading data, concentrations of most VOCs measured by the PTR-MS are significantly higher in the dry season than in the wet season. Concentrations of isoprene and its oxidation products, biogenic precursors of OA, are a factor of 2-3 times higher in the dry season than the wet season. Mean temperatures during the dry season are XX higher and isoprene emissions have been shown to scale with temperature, among other variables [REF Guenther]. Measurements of benzene, which is primarily anthropogenic in origin, were also significantly higher in the dry season. While benzene itself is unlikely to significantly contribute to SOA formation over the timescales we observe on the G-1 flights (4-5 hours) due to its ~5 day atmospheric lifetime, other unmeasured anthropogenic VOC concentrations which would contribute to more immediate OA formation may have been higher as well. The higher measured VOC concentrations may contribute to the higher OA concentrations measured in the dry season. However, the PTR-MS data, in addition to satellite and remote sensing measurements (Martin et al., 2016), also show that biomass burning significantly impacted the region. Measurements of acetonitrile, whose major source is biomass burning (Yokelson et al., 2009;Yokelson et al., 2007), are 2-3 times higher in the dry season than the wet season, indicating a significant biomass burning impact in the region. Biomass burning is a known source of OA (Jolleys et al., 2012;Bond et al., 2004;Yokelson et al., 2009;Ferek et al., 1998) and would also contribute to the higher aerosol concentrations observed in the dry season.

*We have also added additional background information on the wet and dry seasons.*

The timing of these flight missions was chosen to provide a contrast between the wet and dry seasons (Martin et al., 2016). In the wet season, back trajectory analysis indicated that the Manaus region was typically under the influence of air originating from the North Atlantic Ocean (Martin et al., 2016). Regular, organized mesoscale convective systems triggered by sea breeze circulation brought widespread, moderate rate precipitation to the region (Giangrande et al., 2017;Machado et al., 2018;Burleyson et al., 2016). The high level of rainfall and moisture in the wet season inhibited biomass burning and low number fires were observed (Martin et al., 2016;Machado et al., 2018). Under these conditions, the Amazon basin is one of the cleanest continental regions on Earth onto which the Manaus plume represents a significant perturbation (Martin et al., 2010;Artaxo et al., 2013). In the dry season,

back trajectory analysis indicate airmasses originate from the Southern Hemisphere and travel up the Amazon River transporting pollution from the northern coastal cities (Martin et al., 2016). In addition, recirculation events transported air from the southern Amazon basin into the Manaus region (Martin et al., 2016). Intense biomass burning fires in the central and southern portion of the Amazon basin and in central Africa were observed in the dry season and a portion of the emissions were transported to the Manaus region (Martin et al., 2016;Artaxo et al., 2013;Martin et al., 2010). In the dry season, more intense but less frequent convection produced approximately one quarter of the total rainfall observed in the wet season. As a result of the combination of transport and precipitation, the Manaus region is significantly more polluted in the dry season.

More detailed diagnostic information should be presented to confirm the PMF results.

*We have included 4 new figures to the supplementary material and 1 new table in the appendix showing PMF diagnostics. We have also moved our discussion on the PMF factor choices into the appendix and expanded that text.*

[revised manuscript text omitted]

**Figure S3. Reconstruction of the mass of the 5 Factor solution chosen for the wet season data set. The total measure mass is shown in the purple dots and the reconstructed mass is shown as a black line. Individual factors are color coded. To make the information legible, we have chosen to highlight the data for the March 13 flight presented in the main text. Factors 1, 2, and 3 were combined into a single OOA factor when reported in the main text as discussed in the SI above.**

[Figure]

**Figure S4. Mass spectral profiles of the 5 factor solution chosen for the wet season data. Factors 1, 2, and 3 were combined when reported in the main text as discussed in the SI.**

[Figure]

**Figure S5. Plot of the correlation coefficients between the time series and mass spectra of the 5 factor solution.**

For the finding of constant Δorg/ΔCO and increasing aerosol oxidation in an aging urban plumes, one question is the behaviors of VOCs according to PTR-MS measurements?

*Concentrations of isoprene display a non-monotonic dependence on the distance of the plume from Manaus. The average isoprene concentrations are 1.49, 1.36, 0.88, 1.12, and 1.11 ppbv for plume transport distances of 0, 25, 45, 69, and 95 km, respectively. We note that fresh isoprene will continuously mix into the plume as it is transported over the forest. The concentration of m/z 71 (isoprene oxidation products) monotonically increase from 0.96 ppbv directly over the city to 1.27 at 95 km distance. The average tolene:benzene ratio also monotonically decreased with plume age, though conversion of this ratio to a plume age resulted in unrealistic values. We have added this information to the text.*

Concentrations of isoprene are lower in the plume than the background values observed outside of it. Concentrations of isoprene inside the plume do not show a monotonic dependence on plume age, The concentration of isoprene oxidation products measured in the plume by the PTR-MS at m/z 71 monotonically increases with plume aging, from 0.96 ppbv directly over the city to 1.27 ppbv at the farthest leg. The average toluene:benzene ratio in the plume also monotonically decreases with plume aging as would be expected, though attempts to convert these ratios into a photochemical age resulted in unrealistically high estimates of the plume age, likely due to noise in the data as mentioned above (de Gouw et al., 2005).

Was the March 13 flight an isolated case or is the phenomenon more general for the Manaus emissions? What are the Δorg vs ΔCO values for other flights? How do they compare? Are there seasonal differences?

*We have analyzed additional data and discussed this question in more detail in the revised manuscript and added Figure 7 to address it. Please see our response to Reviewer #1 (above)*

*which details the changes made to the manuscript. In short, the Δorg vs ΔCO values for two additional flights are similar to the March 13 data.*

1. Page 5, line 15, what does the "13s data sampling interval" correspond to? Was the AMS operated under the standard MS mode (equal chopper-open and closed positions) or the fast sampling mode typically used for aircraft sampling?

*The AMS was operated in the standard MS mode, with equal chopper open and closed periods. See also our response to a similar comment from Reviewer #1.*

The AMS operated only in the standard "V"- MS mode (the particle sizing mode was not used) with a 13s data sampling interval and equal chopper open and closed periods of 3 seconds.

2. Page 5, line 28, what's the m/z range for the PTR-MS measurement?

*The PTR-MS was run in an ion monitoring mode in which a limited number of pre-selected masses were monitored, rather than a scanning mode where the entire mass spectrum is recorded. In this study, the highest m/z value monitored was 137, corresponding to monoterpenes. We have updated the text to make this more clear. The spectrometer is capable of scanning up to m/z 500.*

The PTR-MS was run in the ion monitoring mode in which the signals of a limited number of pre-selected of m/z values are sequentially measured with one measurement cycle taking 3.5 s.

3. Page 6, line 1, define E/N.

*We have defined E/N in the manuscript.*

Drift tube temperature, pressure and voltage were held at 60 °C, 2.22 hPa, and 600 V, respectively resulting in an electric field to gas density (E/N) ratio of 134 Td (1 Td =$1\times10^{-17}$ $cm^2V^{-1}s^{-1}$).

4. Page 6, what's SAMBBA?

*We have defined SAMBBA in the text.*

…the South American Biomass Burning Analysis (SAMBBA) campaign…

5. Page 6, line 31, mention the year and month for the Chen et al. study.

*The year and month for the Chen study have been added.*

Chen et al. reported AMS-measured wet-season (February 7 to March 13, 2008) campaign average organic and sulfate loadings

6. Page 7, Line 13, are there diurnal mixed layer height data to justify the usage of 700 m?

*As seen in Table 1, the lowest flight legs were flow at 500 m or 600 m altitude above sea level and averaging data below 700 m captures these legs. The 700 m altitude was chosen so that any small altitude corrections due to underlying terrain or any small error in the altitude reading did not exclude data on the lowest altitude legs. Radiosonde balloons were launched from the T3 site*

*5 times a day every day during the campaign. The average diurnal mixed layer heights for both the wet and dry season GoAmazon 2014/15 periods based on these radiosondes are shown in Figure 3 of Giagrande et al. (2017) (Giangrande et al., 2017) They report average mixed layer heights of greater than 1000m above ground level by 10:00 local time in both the wet and dry seasons. The Manaus elevation is ~100 m, so yes, on average we were sampling the boundary layer at 700 m altitude during the flights. We have added a this information to the manuscript.*

Giagrande et al. (2017) analyzed radiosonde data from the T3 site and report average mixed layer heights of greater than 1000 m above ground level at 10:00 local time in both the wet and dry seasons. Thus, the G-1 was typically sampling in the boundary layer on the lowest flight legs and the 700 m data should represent boundary layer conditions.

Page 7, line 18, what does 'sources' mean in this context?

*We are referring to the influence of biomass burning in the dry season. We have revised this sentence to be more clear.*

The fractional contribution of each species to the total loading is nearly identical when comparing seasons, despite the large differences in aerosol absolute mass concentrations and the larger influence of biomass burning in the dry season (Martin, 2016).

8. Page 7, line 22, what does "aircraft product distribution' mean?

*Aircraft product distribution means averaged fractional composition of the aerosol measured from the aircraft. We have revised this sentence to be more clear.*

The fractional contributions of each chemical species described herein are nearly identical to the previous reports from Chen et al. (2009, 2015).

9. Page 7 line 24, "suggests"

*Corrected.*

10. Page 7, line 30 - 31, was biomass burning influence detected during this study? What are the evidences? The citations were clearly not from Go-Amazon 2014/15.

*Yes, biomass burning influence was detected through remote sensing, both by satellite and ground-based instruments as described in Martin 2016. We have added that reference to lines 30-31 and throughout the manuscript. In addition, the PTR-MS measurements of acetonitrile suggest a significant biomass burning influence. We have added this information to the text. Finally, the biomass burning background was often readily observed by the naked eye both from the aircraft and from the ground. The haze observed visually in the dry season was in stark contrast the clear conditions observed in the wet season.*

In the dry season, back trajectory analysis indicate air masses originate from the Southern Hemisphere and travel up the Amazon River transporting pollution from the northern coastal cities (Martin et al., 2016). In addition, recirculation events transported air from the southern Amazon basin into the Manaus region (Martin et al., 2016). Intense biomass burning fires in the central and southern portion of the Amazon basin and in central Africa were observed in the dry season and a portion of these emissions

were transported to the Manaus region (Martin et al., 2016;Artaxo et al., 2013;Martin et al., 2010). In the dry season, more intense but less frequent convection produced approximately one quarter of the total rainfall observed in the wet season. As a result of the combination of transport and precipitation frequency, the Manaus region is significantly more polluted in the dry season.

However, the PTR-MS data, in addition to satellite and remote sensing measurements (Martin et al., 2016), also show that biomass burning significantly impacted the region. Measurements of acetonitrile, whose major source is biomass burning (Yokelson et al., 2009;Yokelson et al., 2007), are 2-3 times higher in the dry season than the wet season, indicating a significant biomass burning impact in the region. Biomass burning is a known source of OA (Jolleys et al., 2012;Bond et al., 2004;Yokelson et al., 2009;Ferek et al., 1998) and would also contribute to the higher aerosol concentrations observed in the dry season.

11. Page 7, line 32, spell out MSA.

*We have spelled out MSA.*

12. Page 8, line 1-2, give citation.

*We were discussing our own observations, but have decided to delete this sentence as it was not illustrated in the figures and thus confusing.*

13. How well is the correlation between IEPOX-SOA and m/z 82?

*The correlation between the IEPOX SOA factor and the $C_5H_6O$ ion has a Pearson correlation coefficient of r=0.80 for the entire wet season dataset.*

14. Page 10, line 5, "this date"

*We have fixed this error.*

15. Page 10, line 15 – 16, might be useful to report the correlation coefficients.

*We have added the correlation coefficients. As discussed in detail in the manuscript, the slope of these correlations changes with plume age. As a result, the correlations are degraded by the changing slope and correlation coefficients for individual legs are larger.*

The HOA factor correlates strongly with CO (r=0.79 for all data at 500 m) while the OOA factor correlates strongly with ozone (r=0.81 for all data at 500 m), though we will show below that the slope of the correlation changes with plume age. The changing slope has the effect of degrading the HOA/CO and OOA/ozone correlations which are larger for individual plume legs.

16. Page 10, line 18, clarify the meaning of "transformation of HOA to SOA/OOA", e.g., through what mechanisms. Is oxidized POA counted as SOA?

*As discussed later in the text (e.g., page 12 lines, 20-26) we are unsure of the chemical mechanism. The previous literature observations have generally attributed this conversion to evaporation of HOA, oxidation of the volatilized HOA in the gas phase, and re-condensation of the oxidized HOA as SOA (Robinson et al., 2007;Sage et al., 2008). In this mechanism, oxidized POA would be counted as SOA. Alternative possibilities leading to loss of HOA include dry*

*deposition and evaporation of HOA without oxidation and re-deposition of SOA. In this case, SOA deposition could come from either anthropogenic VOCs emitted in the Manaus plume or from oxidation of biogenic VOCs in the plume. We are unable to differentiate these mechanism sufficiently and we therefore tried to write the indicated text so as not to rule out any of these possibilities.*

17. Page 10, line 23 - 24, this sentence is vague. What chemical mechanism?

*We refer to heterogeneous uptake of gas-phase IEPOX on pre-existing aerosol surfaces to form IEPOX-derived SOA. We have revised the sentence to be more clear.*

The IEPOX SOA factor does not appear to change with passage through the plume and does not correlate with sulfate aerosol on this day as may be expected based on the chemical mechanism, which requires heterogeneous uptake of gas-phase IEPOX onto inorganic aerosol to generate IEPOX SOA…

18. Page 10, line 26, limit of detection for IEPOX SOA was mentioned. What's the value and how was it determined?

*The limit of detection for organics in this study, based on analysis of filter measurements as described in DeCarlo et al. (2006), is approximately 0.12 ug/m3, for the 13s averaging time we utilized. We have added this information to the manuscript (see also our response to a question on detection limits by reviewer #1). In addition, Ulbrich et al. (2009) report that factors that are less than 5% of the total mass are not accurately retrieved by the PMF routine. On the 500 m flight leg for the March 13 flight, neither criteria are met. We note that on other flights, IEPOX-SOA is above these detection limits.*

Based on the standard deviation of these blank measurements (3σ) as described in the literature (DeCarlo et al., 2006), the detection limit of the AMS at the 13s sampling interval were 0.12, 0.01, 0.014, and 0.005 µg/m$^3$ for organics, sulfate, nitrate, and ammonium respectively.

19. Page 11, line 5 -9, how was Δorg/ΔCO determined, through background subtraction or linear fit? Either way, please provide details, e.g., the choice of background values or quality of linear fit etc. Consider to move some information in the supplementary to main body of the manuscript.

*We use method 1 (linear fit with no background subtraction) for data shown in the main manuscript. However, it is important to note that, as we show in the supplementary material, results are insensitive to the chosen method and to reasonable choices of background CO and org values. We have moved some information on the Δorg/ΔCO calculate from the SI to the main text.*

The Δorg/ΔCO values shown in Figure 7 were calculated as the slope of a linear regression line between the AMS-measured organic mass loadings and CO concentrations. All data for each flight leg perpendicular to the wind direction were included. Background values of OA and CO were not subtracted from the data and the regression was not forced through the origin. We acknowledge that calculation of Δorg/ΔCO values can be sensitive to the calculation method and choice of background values for both OA and CO; therefore, we performed calculations using different methods and assuming

different background concentrations for both OA and CO and find general agreement between these methods (see SI for more details).

20. Page 12, line 14 – 15, change "occurring" to "is occurring".

*We have corrected this error.*

21. Page 13, line 18, speaking of sulfur emissions and oxidation of compounds such as DMS, it would be necessary to check for the presence of methylate /MSA in particles. BTW, DMS should be defined.

*We have defined DMS.*

*The most recent literature method for identifying MSA in aerosol involve utilizing the $CH_3SO_2,^+$ ion as a signature (Huang et al., 2017). Due to its mass defect, this ion is well separated from other typical aerosol components so it should be easily detectable if present. We do not detect evidence of this ion in our dataset. Thus, we can't positively identify MSA in the particles. The oxidation of DMS and $H_2S$ was described in other literature studies as significant sources of sulfate in the Amazon Basin, though not the major source (Andreae et al., 1990;Chen et al., 2009). We are unsure if this represents a discrepancy or not. One possible explanation is that $H_2S$, which was found in approximately 10x the concentration of DMS, is responsible for most of the biogenic sulfate production (Andreae et al., 1990). Our background measurements are roughly consistent with the literature estimate of the natural in-basin contribution to sulfate aerosol.*

22. Page 14, line 5, it is not that aerosol composition "did not change" but rather "did not change significantly".

*We have made the suggested edit.*

23. Figure 1, consider to mention in the figure caption the duration of the flight.

*We have added the flight time to the figure caption. In response to a comment from Reviewer 1, we have also added a table with the flight times for all research flights.*

Takeoff time was 10:14 (local time) and landing time was 13:21 for a total flight duration of three hours and nine minutes.

24. Figure 2, how were the average pie charts calculated, straight averages of the fractions or mass weighted averages?

*The Figure 2 pie charts were calculated by binning all the mass loading data from a particular season together and calculating the fractions. We have added this clarification to the text.*

The bottom panel shows the distribution of the chemical species as a mass weighted average of all data from the respective measurement season.

25. Figure 4, What are the aerosol species concentrations, same as mentioned in Fig. 2 (23C and 1 atm)?

*All data presented in the manuscript are normalized to 23 C, 1 atm. We have added a note to all figure captions mentioning the normalization. We have also pointed out in the experimental section that data presented in the manuscript are normalized to 23 C, 1 atm.*

26. According to Figure 4, the time series of O3 and CO appear to correlate quite well. Figure 5 shows that HOA and OOA go up and down together as well. It would be interesting to know the cross correlations between OOA, HOA, CO and O3.

*A cross correlation table listing the Pearson correlation coefficient (r) for each species is shown below. In general, ozone and CO are correlated but the slope of the correlation changes as ozone is formed with plume aging. This correlation breaks down very close the city, as seen in figure 4 around 11:45 where ozone and CO are briefly anti-correlated. On this portion of the flight, the correlation of HOA and OOA also breaks down.*

|       | OOA  | HOA  | Ozone | CO   |
|-------|------|------|-------|------|
| OOA   |      | 0.64 | 0.81  | 0.77 |
| HOA   | 0.64 |      | 0.58  | 0.79 |
| Ozone | 0.81 | 0.58 |       | 0.71 |
| CO    | 0.77 | 0.79 | 0.71  |      |

27. Figure 5, was NO2 measured? How does Ox time series look like?

*NO$_2$ measurements were attempted, but exposure of the instrument to high NOx concentrations on the airport tarmac lead to artificially high NO$_2$ measurements, particularly early in the flights. The figure below shows the ozone, Ox and NO$_2$ trace appended to a simplified version of Figure 5. Also below is a table presenting the mean ozone and Ox measured in the plume as a function of leg number and hence plume aging. The figure shows that Ox is dominated by ozone, though NO2 may be a significant fraction of Ox close to the city (e.g. around 11:45). The table shows that these quantities increase from Legs 1-4 and decrease on Leg 5. Trends in peak ozone/Ox show a similar pattern. Due to the bias in the NO$_2$ measurement, we prefer not to publish the Ox time series in the final manuscript.*

[Figure]

| | Distance (km) | mean NO$_2$ (ppb) | mean O$_3$ (ppbv) | mean O$_x$ (ppbv) |
|---|---|---|---|---|
| Leg 1 | 0 | 0.77 | 12.14 | 12.89 |
| Leg 2 | 24 | 1.49 | 16.27 | 17.73 |
| Leg 3 | 45 | 2.29 | 20.84 | 23.07 |
| Leg 4 | 69 | 1.31 | 25.96 | 27.54 |
| Leg 5 | 95 | 0.69 | 21.95 | 22.63 |

28. Figure 6, consider to add error bars for the data.

*We have added error bars to the data. The error bars are based on standard addition of errors of the AMS and CO measurements. To keep the graph legible and clear, we added an error bar to only one point in each measurement series.*

[Figure]

**References**

[revised manuscript text omitted]